# Preliminary bone histological analysis of *Lystrosaurus* (Therapsida: Dicynodontia) from the Lower Triassic of North China, and its implication for lifestyle and environments after the end-Permian extinction

**Fenglu Han**[1]*, **Qi Zhao**[2], **Jun Liu**[2,3,4]*

**1** School of Earth Sciences, China University of Geosciences (Wuhan), Wuhan, China, **2** Key Laboratory of Vertebrate Evolution and Human Origins of Chinese Academy of Sciences, Institute of Vertebrate Paleontology and Paleoanthropology, Chinese Academy of Sciences, Beijing, China, **3** CAS Center for Excellence in Life and Paleoenvironment, Beijing, China, **4** College of Earth and Planetary Sciences, University of Chinese Academy of Sciences, Beijing, China

* hanfl@cug.edu.cn (FH); liujun@ivpp.ac.cn (JL)

**Data Availability Statement:** All relevant data are within the manuscript and its Supporting Information files.

## Abstract

*Lystrosaurus* represents one of the most successful dicynodonts, a survivor of the end-Permian mass extinction that remained abundant in the Early Triassic, but many aspects of its paleobiology are still controversial. The bone histology of *Lystrosaurus* species from South Africa and India has provided important information on their growth strategy and lifestyle, but until recently no data was available on the bone histology of *Lystrosaurus* from China. Here, we report on the bone microstructure of seven *Lystrosaurus* individuals from the Lower Triassic of Xinjiang, providing the first such data for the Chinese *Lystrosaurus* species. Our samples indicate that the microstructure of *Lystrosaurus* limb bones from China is characterized by fibrolamellar bone tissue similar to those from South Africa and India. Three ontogenetic stages were identified: juvenile, early subadult, and late subadult based on lines of arrested growth (LAGs) and bone tissue changes. Bone histology supports a rapid growth strategy for *Lystrosaurus* during early ontogeny. Unlike Early Triassic *Lystrosaurus* from South Africa, lines of arrested growth are common in our specimens, suggesting that many individuals of Chinese *Lystrosaurus* had reached the subadult stage and were interrupted in growth. The differences in bone histology between *Lystrosaurus* from South Africa and China may indicate different environmental conditions in these two regions.

## Introduction

The extinction at the end of the Permian was the most influential event in animal history since the Cambrian explosion. About 80% of all marine species and 50–70% of terrestrial families went extinct [1–3]. Recent studies suggest that the Permian-Triassic transition occurred in an

**Funding:** This study was supported by Strategic Priority Research Program of Chinese Academy of Sciences (XDB26000000) to JL, the National Natural Science Foundation of China (www.nsfc. gov.cn; 41530104, 41661134047) to FH and JL respectively. The funders had no role in study design, data collection and analysis, decision to publish, or preparation of the manuscript.

**Competing interests:** The authors have declared that no competing interests exist.

increasingly hot environment which was disruptive to life [4–8], although it is still controversial whether the environment was arid [6, 7] or humid [9]. However, some organisms evolved new survival strategies, such as miniaturization or rapid propagation [10–12].

*Lystrosaurus* is a medium to large-sized dicynodont therapsid with a strongly anteroventrally downturned snout. It represents one of the most successful vertebrates that survived the end-Permian mass extinction, showing a cosmopolitan distribution during the Early Triassic [13–17]. However, the survival strategies of *Lystrosaurus* during the extinction are still controversial [18, 19]. The lifestyle of *Lystrosaurus* was proposed as being aquatic or semi-aquatic, which could have helped it to escape the harsh terrestrial environment [20, 21], and was supported by histological studies [22, 23] (but also see Botha et al. [24]). There is also evidence that *Lystrosaurus* was a burrow maker [25, 26], and burrowing can help an organism survive harsh environmental conditions, as temperature and humidity are generally more stable underground than on the surface [27, 28]. Also, underground burrows can provide better protection for the young, which helps propagate the species [29].

Interestingly, the body size of *Lystrosaurus* from South Africa is smaller in the Early Triassic than in the late Permian [18], suggesting miniaturization. Furthermore, LAGs (lines of arrested growth) are more common in Permian *Lystrosaurus*, which was interpreted as evidence that the Permian species had a more extended ontogeny extended over several years [19]. This contrasts with the prevalence of small individuals with few or no LAGs in the Triassic species, which was part of the evidence suggesting early breeding to compensate for dying at an early age [19].

Bone histology can provide important information on the life history of *Lystrosaurus*, which may bear on its survival during the end-Permian mass extinction. Previous studies have focused on the *Lystrosaurus* record from South Africa and India [22, 23, 30, 31]. However, *Lystrosaurus* seems to have been cosmopolitan in distribution [14], and is also very abundant in the Juggar and Turpan basins of Xinjiang, China. In China, seven *Lystrosaurus* species have been named since 1935, but the validity of most of these needs re-evaluation [32, 33], and it is possible that only two species (*L. hedini* and *L. youngi*) are valid [34]. Here, we present data on the bone histology of *Lystrosaurus* from the Lower Triassic Turpan Basin, Xinjiang Province, China for the first time, using this data to comment on the survival strategy of this taxon.

## Material and methods

Seven specimens of *Lystrosaurus* from the collections of the Institute of Vertebrate Paleontology and Paleoanthropology (IVPP) were selected for histological analysis. All materials come from the Lower Triassic Jiucaiyuan Formation of the Taodonggou area (Taoshuyuan), Turpan Basin, Xinjiang, China [35, 36]. The age of the Jiucaiyuan Formation in this area is middle-upper Induan based on previous chronostratigraphy [35, 37]. Identification of these specimens to species level is currently not possible because the Chinese *Lystrosaurus* taxonomy is currently based on skull characters.

Sections were sampled from the mid-diaphysis of limb bones including the tibia, femur, humerus, ulna, and radius (S1 Fig). However, IVPP V26546 was sampled near the distal end (the distance between the sampled position and the distal end is about 5 cm) of the femur due to the lack of a mid-region. Limb diaphyseal sections were chosen because they are the primary centers of ossification and should preserve the most complete record of growth, and also minimize the potential of secondary remodeling overwriting the growth record (S1 Fig; Table 1) [38, 39]. Rib sections in some individuals were also sampled for comparison. All these materials were photographed and measured before sectioning (S1 Fig; Table 1). However, the sampled specimens preserved different parts of limb bones making the comparison of body size difficult. Therefore, we estimated values of missing limb bones based on comparisons with a

**Table 1. Skeletal elements of the *Lystrosaurus* specimens examined for histological sampling from the Early Triassic of Xinjiang, China.**

| Specimen number | Sampled elements | Humeral length (cm) | Ulna length | Radial length (cm) | Femoral length (cm) | Tibial length (cm) | Length of fibulae (cm) | Growth stages |
|---|---|---|---|---|---|---|---|---|
| IVPP V26543 | Tibia | 9.9[e] | 8.3[e] | 7.4[e] | 12.5 | 9.32 | 9.0 | Juvenile |
| IVPP V26544 | Humerus, Tibia, Fibula | 10.1 | 7.8[e] | 7.0[e] | 11.8 | 9.17 | 9.0 | Juvenile |
| IVPP V26542 | Femur | 10.3 | 8.3[e] | 6.52 | 12.5[e] | 9.8[e] | 9.5[e] | Juvenile |
| IVPP V26545 | Fibula, Radius | 15.0 | 11.0 | 10.8[e] | 18[e] | 14.2[e] | 13.8[e] | Early subadult |
| IVPP V26546 | Femur | 12.0 | 9.7[e] | 8.7[e] | 14.6[e] | 12.5 | 11.5 | Late subadult |
| IVPP V26547 | Femur | 12.4[e] | 10[e] | 9[e] | 15.1 | 12.1 | 11.3 | Late subadult |
| IVPP V26548 | Rib | 15.6[e] | 12.6[e] | 11.3[e] | 19.0 | 14.0 | 13.5 | Late subadult |
| IVPP RV35012 | None | 11.5 | 9.3 | 8.3 | 14.0 | 10.9 | 10.6 | Subadult |

[e] denotes estimated values based on the comparison of the ratios between elements of an almost complete skeleton of *Lystrosaurus hedini* (IVPP RV35012). The estimated values in each element are all compared to the femur, whereas there is no femur preserved, the estimated values are compared to the humerus. "Growth stages" of all individuals except IVPP RV35012 were identified based on bone microstructure.

complete skeleton of *Lystrosaurus hedini* (IVPP RV35012) [40]. It has a skull length of 173 mm and includes most of the postcranial material. It is probably a subadult individual based on open neurocentral sutures and fused sacral vertebrae. Comparison of the limb bone length (Table 1) with other individuals also ascribes it to be within a subadult stage.

The preparation of the histological sections was carried out at the IVPP. The selected bones were processed using the Exakt-Cutting-Grinding System [41]. Sections were embedded in polyester resin, cut by EXAKT-300CP, and were ground to a thickness of about 50 μm with an EXAKT 400CS grinding machine. The polished sections were then observed under transmitted and polarized light. The thin sections were observed using a Leica DM2700P microscope and photographed using a Leica DMC4500 camera. The whole cross-sections are large and bone microstructure varies in different areas around the cortex, and we have labeled all the sampled directions for all the figures for easy comparison. Nomenclature and definitions of structures follow Francillon-Viellot and Chinsamy-Turan [42, 43]. 'LAGs' are growth mark lines that can be traced around the full circumference, whereas growth marks are only partially shown and are not circumferential [38, 44]. A LAG is a cement line and indicates a temporary but complete cessation in growth that represents one year, whereas it is uncertain whether a growth mark indicates seasonal or yearly patterns. Here we assume that only LAGs to be an indicator of yearly patterns. The annuli correspond to periods of slow growth by the presence of parallel-fibered bone and flattened osteons [42].

The cortical thickness was calculated as the ratio of the cortex (one side) to the cross-sectional diameter of the bone (S1 Table) [22]. The cortex can be identified by bone tissue being connected tightly forming net and having no bone trabecula and large resorbed cavity, whereas the medullary cavity is either empty (free medullary cavity) or filled with trabeculae, which are usually isolated or loosely connected. However, sometimes, the trabeculae are abundant and show a similar connection as in compact bone, and could not be separated clearly in this situation, and then the cortical thickness could not be provided. In addition, cortical thickness usually varies in different parts of the whole cross-section, and we have provided a range of cortical thickness measurements in each section (Table 2). The measurement positions of the maximum and minimum of the cortical thickness in available sections are shown in S2 Fig.

**Table 2. Bone histological information of *Lystrosaurus* from China in each specimen.**

| Specimen number | Skeletal elements | Cortical thickness (%) | Cortical porosity (%) | Bone tissue type | Secondary reconstruction | Growth rings | Medullary cavity |
|---|---|---|---|---|---|---|---|
| IVPP V26543 | Tibia | 24–32 | 29 | plexiform and reticular | inner enlarged channels | 0 | a small free cavity |
| IVPP V26544 | Humerus | 21–45 | 32 | plexiform | a few secondary osteons | 0 | a small free cavity |
| IVPP V26544 | Tibia | 19–30 | 27 | laminar and reticular | a few secondary osteons | 0 | full of trabeculae |
| IVPP V26544 | Fibula | / | / | laminar | a few secondary osteons | 0 | a large free cavity |
| IVPP V26542 | Femur | 16–41 | / | laminar and plexiform | a few secondary osteons | 0 | a free medullary cavity |
| IVPP V26545 | Fibula | / | / | laminar | a few secondary osteons | 1 | full of trabeculae |
| IVPP V26545 | Radius | / | / | laminar | a few secondary osteons | 1 | full of trabeculae |
| IVPP V26546 | femur | / | / | laminar | extensive secondary reconstruction | 3 | full of trabeculae |
| IVPP V26547 | Femur | / | / | laminar | extensive secondary reconstruction | 4 | full of trabeculae |
| IVPP V26548 | rib | / | / | laminar | extensive secondary reconstruction | >2 | full of trabeculae |

Cortical porosity was measured using the software ImageJ 1.52a (https://imagej.nih.gov/ij/). It was difficult to obtain the cortical porosity of a whole cross-section for each specimen because of partial breakage or poor preservation in some areas of these sections. Therefore, we subsampled ten areas from the inner to the outermost cortex in all sections to calculate the average vascular porosity (S1 Table). The sampled areas were chosen from the inner region to the periphery on different parts of the whole cross-sections (S3 Fig), so they could reflect the value of cortical porosity. However, some bone sections show unclear bone histology in some regions, which may affect the results. Cortical porosities of those bone sections that were poorly preserved or with an unclear transition between compact bone and medullary cavity are not provided. High-resolution images were acquired using a Leica DMC4500 camera. All the images were converted to 8-bit files, and the brightness/contrast was adjusted to clarify the structure. After that, we used "threshold" tools in ImageJ to select all the vascular canals. Finally, the cortical porosity was determined using "analyze particles" tools in ImageJ.

## Results

The bone histology of *Lystrosaurus* from North China shows a predominance of fibrolamellar bone tissue and a high cortical porosity (Figs 1–6), as in *Lystrosaurus* from South Africa and India [22, 24]. The primary osteons are predominantly longitudinal, forming laminar or plexiform bone. Radially oriented primary osteons are also present in some parts of the cortex. Secondary remodeling is common in the inner and mid-cortical bone in the subadult stage, resulting in enlarged channels and thin periosteal bone. The medullary cavity is small and filled with bony trabeculae. LAGs are common in the outer cortex of larger individuals.

All these elements could be divided into juvenile, early subadult and late subadult stages based on bone tissue type and the presence of LAGs (Table 2). They are generally similar to the growth stages of *Lystrosaurus* from South Africa and India, but also have some features that differ from the latter.

### Juvenile

The juvenile stage was represented by three samples: IVPP V 26543, VPP V 26544 and IVPP V26542. All the cortices are composed of fibrolamellar tissue, with a high number of vascular

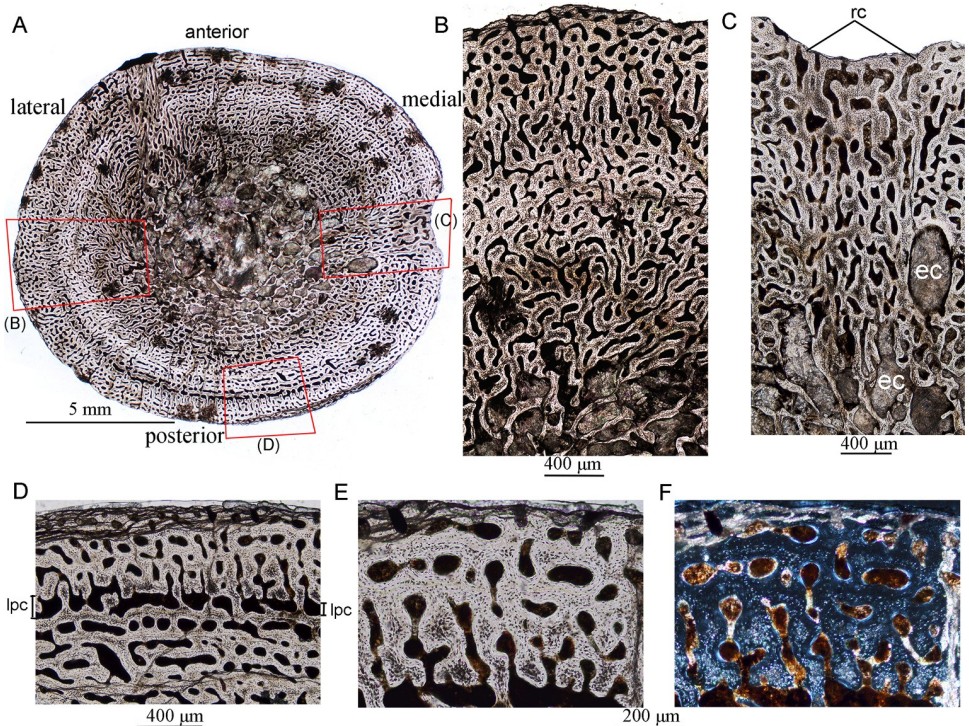

**Fig 1. Bone microstructure in juvenile *Lystrosaurus* IVPP V26543. A**, whole cross-section of the tibia; **B**, partial cross-section showing fibrolamellar bone; **C**, partial cross-section showing radial vascular canals; **D**, the outer cortex under normal light; **E**, enlargement of the outer cortex showing primary osteons under normal light; **F**, bone tissue of (E) under polarized light. **Abbreviation**: **ec**, enlarged resorption cavity; **rc**, radial canals; **lpc**, large primary canals.

canals, abundant globular osteocyte lacunae, and no LAGs, suggesting a rapid, constant growth rate. The mid-diaphyseal tibia of IVPP V 26543 is elliptical in cross-section, with a diameter of 13.5 (ML, mediolateral width) × 12.0 (AP, anteroposterior width) mm and a cortical thickness ranging from 24–32% (Fig 1; Table 2). The free medullary cavity is small, with a diameter of about 4 mm. The entire cortex is comprised of fibrolamellar bone and dense vascular canals (Fig 1). The cortex has a high cortical porosity (about 29%, Table 2). The canals are predominantly longitudinal and arranged in circumferential rows in parts. A few irregularly shaped canals are shown in the inner and mid cortex (Fig 1B). Radial anastomoses are shown in the medial and anterolateral regions of the cortex, and the medial region has a concave periosteal surface (Fig 1A and 1C). The anastomoses join two or three adjacent canals in the outer cortex (Fig 1D) and link four to five canals forming long canals in the mid and inner cortex (Fig 1B). Most of the vascular channels show thin centripetal osteonal deposits forming primary osteons (Fig 1E and 1F). The primary osteons mainly show laminar and plexiform patterns in the outer and deeper cortex, and a row of large primary canals are shown near the periphery (periosteal surface) in some areas (Fig 1D). Osteocyte density is consistently high throughout the cortex. Erosionally enlarged resorption cavities are present in the inner cortex (Fig 1C). There are no secondary osteons, LAGs, or endosteal bone. The medullary region contains cancellous bone.

In IVPP V26544, the tibia, humerus, and fibula were all sectioned (Fig 2). The cross-section of the tibia is elliptical in outline, with a diameter of 12.5 (ML) × 10.0 (AP) mm (Fig 2A). The cortical thickness ranges from 19% to 30% (Table 2). There is no free medullary cavity. The cortex mainly consists of fibrolamellar bone tissues (Fig 2B), but parallel-fibered bone tissue is

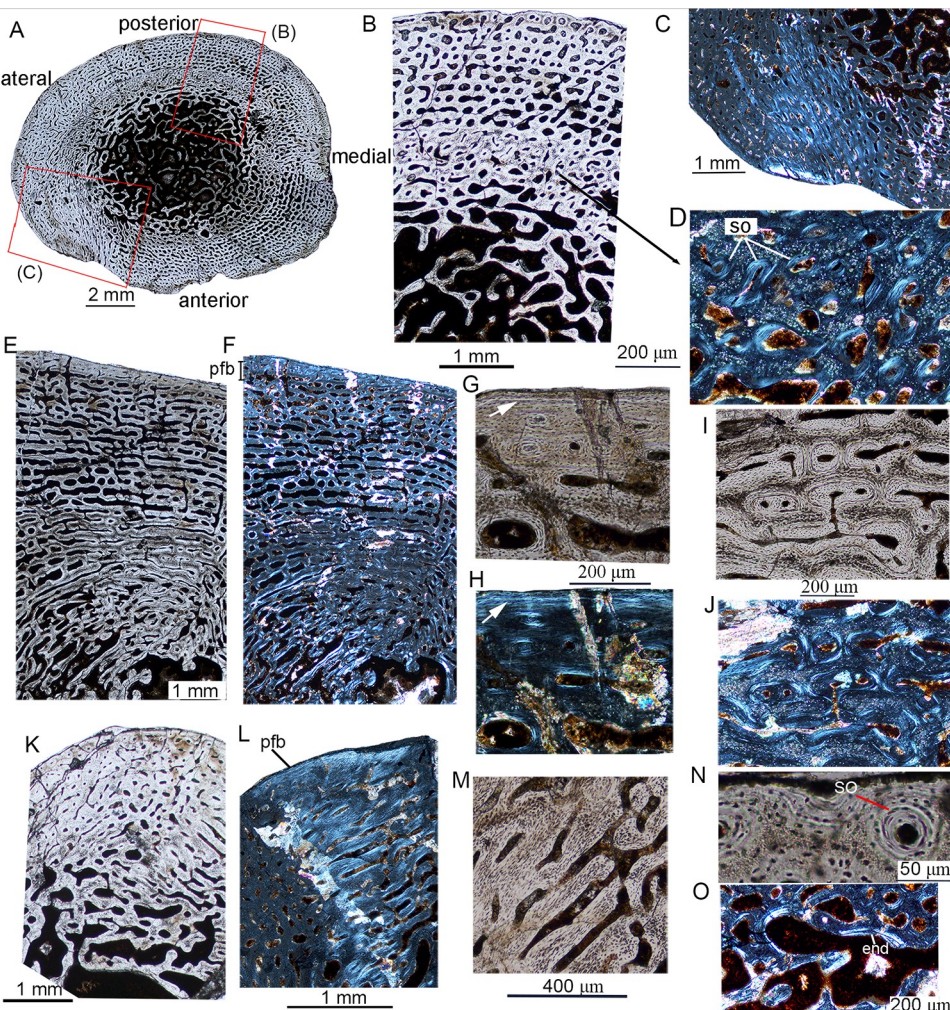

**Fig 2. Bone microstructure in juvenile *Lystrosaurus* IVPP V26544.** (A-D) tibia. **A**, mid-diaphyseal cross-section of the tibia. **B**, bone microstructure of the tibia; **C**, partial cross-section of the tibia showing the parallel-fibered bone under polarized light; **D**, enlargement of the inner region of (B) under crossed plane-polarized light showing secondary osteons; (E-J) humerus. **E**, bone microstructure of the humerus under normal light; **F**, bone microstructure of the humerus under polarized light; **G**, the outermost cortex under normal light showing a growth mark (arrow); **H**, the same region as (G) under polarized light showing parallel-fibered bone; **I**, the inner cortex showing secondary osteons under normal light; **J**, the same region as (I) under polarized light; (K-O) fibula. **K**, bone microstructure of the fibula under normal light; **L**, bone microstructure of the fibula under polarized light showing parallel-fibered bone; **M**, enlargement of (L) showing the organized osteocyte lacunae; **N**, the outermost cortex showing secondary osteons under normal light; **O**, the innermost region showing endosteal bone under polarized light. **Abbreviation**: **end**, endosteal bone; **so**, secondary osteons.

present in the medial region of the cortex that articulates with the fibula (Fig 2C). Vascular canals are predominantly longitudinal. There are a few anastomoses that link two to five canals forming long straight and irregular canals. Most of the vascular canals are primary osteons forming a laminar pattern (Fig 2B). A thin layer (1 mm) of secondary osteons is present in the inner region of the compact bone (Fig 2D). The cortex has a high porosity of 27% (Fig 2A, Table 2), but the diameter of vascular canals is slightly reduced near the periphery. Enlarged vascular canals are present in the inner cortex due to secondary remodeling. No LAG/annulus was detected in the cortex. The medullary region is full of trabeculae.

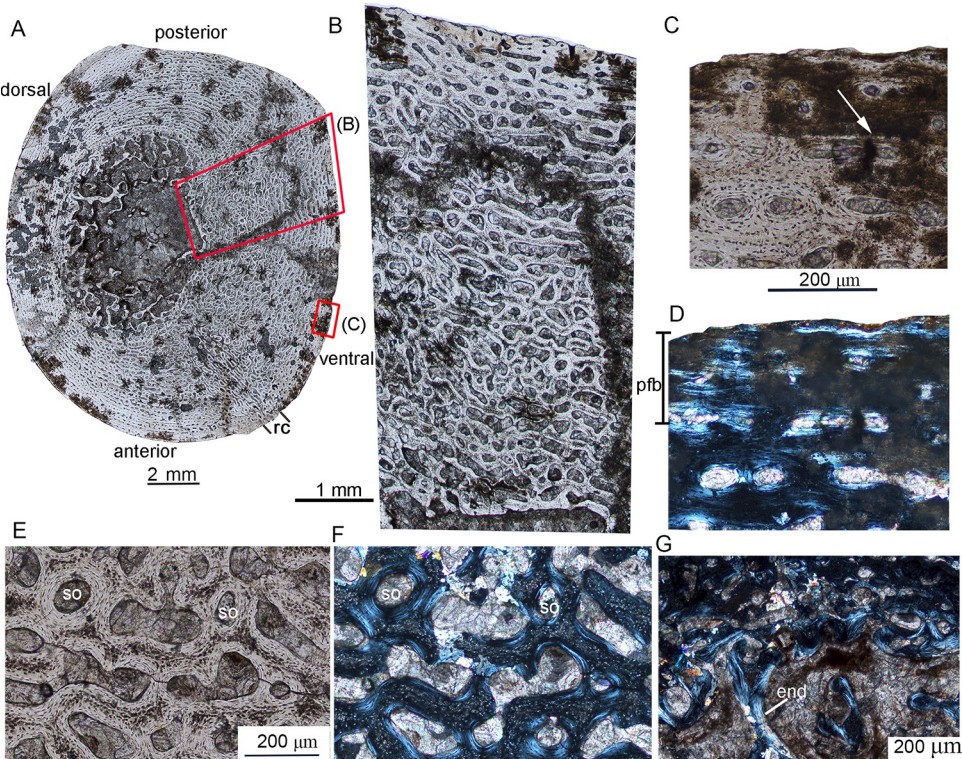

**Fig 3. Bone microstructure of the femur in juvenile *Lystrosaurus* IVPP V26542. A**, mid-diaphyseal transverse section of the femur; **B**, enlargement of (A) showing highly vascularized cortical porosity; **C**, enlargement of the outer cortex showing a growth mark (arrow) under normal light; **D**, the same region as (C), showing parallel-fibered bone under polarized light; **E**, the inner cortex showing secondary osteons under normal light; **F**, the same region as (E) under polarized light; **G**, the medullary region showing endosteal bone. **Abbreviation**: **end**, endosteal bone; **rc**, radial canals; **so**, secondary osteons.

The humerus is subtriangular in cross-section with a midshaft diameter of 18.5 (AP) × 15.5 (DV, dorsoventral width) mm and has a greater cortical thickness than the tibia (Fig 2E and 2F and S3 Fig). It has a thick compact bone (at least 7 mm) with a cortical thickness ranging from 21% to 45% (Table 2), and the free medullary cavity is extremely small. The cortex is comprised of fibrolamellar bone tissue, but a thin layer of parallel-fibered bone tissue is present near the periphery (Fig 2F–2H). The bone porosity has a value of about 32% (Table 2), which is higher than the tibia. The primary osteons are predominantly plexiform and are longitudinally-oriented in the outer cortex. They are strongly reduced in size in the outer cortex. Parallel-fibered bone tissue and one growth mark are present in the outermost region of the cortex (Fig 2G and 2H), suggesting slower growth rates when the individual died, but the osteocyte lacunae are still abundant. Many secondary osteons are seen in the inner cortex (Fig 2I and 2J). A thin layer of lamellar endosteal bone is deposited along the edges of the trabeculae in the medullary region (Fig 2F).

The cross-section of the fibula is oval in outline (midshaft diameter 6.5 × 4 mm, S3 Fig) with a midshaft diameter of 6.3 (AP) × 3.8 (ML) mm. The medullary cavity is small with a diameter of about 2 mm. The cortex mainly consists of fibrolamellar bone tissue, but parallel-fibered bone tissue is shown in the posterior cortex (Fig 2K and 2L). The cortex is less vascularized than the tibia and humerus. Osteocyte lacunae are abundant but are flattened and organized in parallel-fibered bone (Fig 2M). The primary osteons are predominantly longitudinal

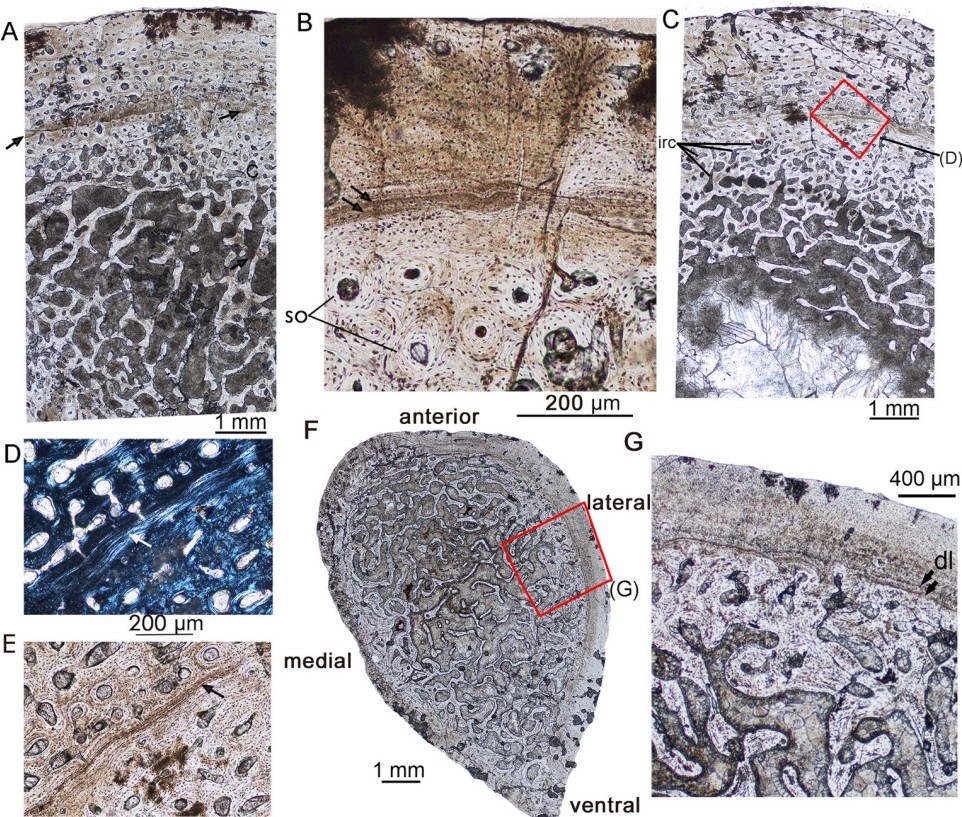

**Fig 4. Bone microstructure in early subadult *Lystrosaurus* IVPP V26545. A**, mid-diaphyseal transverse section of the fibula under normal light. The arrow denotes a LAG. **B**, partial outer cortex of the fibula showing two closely spaced rest lines (arrows). **C**, partial outer cortex of the radius under normal light. **D**, mid-cortex showing a distinct LAG under polarized light (arrows); **E**, the same region as (D) under normal light. **F**, the whole cross-section of the rib under normal light; **G,** enlargement of the outer cortex of the rib. The black arrows indicate a double LAG. **Abbreviation**: **dl**, double LAG; **irc**, irregular canals; **so**, secondary osteons.

and gradually decrease in size towards the periphery. Secondary osteons are mainly in the inner region, but there are also a few near the periphery (Fig 2N). A thin layer of endosteal bone is deposited in the innermost cortex (Fig 2O).

The cross-section of the femur of IVPP V26542 is elliptical with a diameter of 15.0 (AP) × 12.5 (DV) mm and a cortical thickness ranging from 16% to 41% (Fig 3). The estimated bone porosity of the whole thin section is about 38%. The medullary cavity is relatively large (the diameter is about 5 mm) compared to those of juvenile individuals. The thickness of the cortex differs around the bone, and the dorsal cortex is much thinner than the ventral cortex (Fig 3A). Furthermore, the thickened ventral cortex has a higher porosity (about 40%) than that of the narrow dorsal cortex (about 17%). The cortex mainly consists of fibrolamellar bone and is highly vascularized. The primary osteons show a laminar pattern that is mainly longitudinally oriented, but radially-oriented channels are found in a narrow channel near the anteroventral cortex (Fig 3A). The number and diameter of the vascular canals are reduced near the periphery. The outer cortex consists of a thin layer of parallel-fibered bone matrix and a growth mark (Fig 3C and 3D), suggesting slightly reduced growth rates before it died. The secondary osteons are restricted to the innermost cortex (Fig 3E and 3F). Endosteal bone is present along the trabeculae in the medullary cavity (Fig 3G).

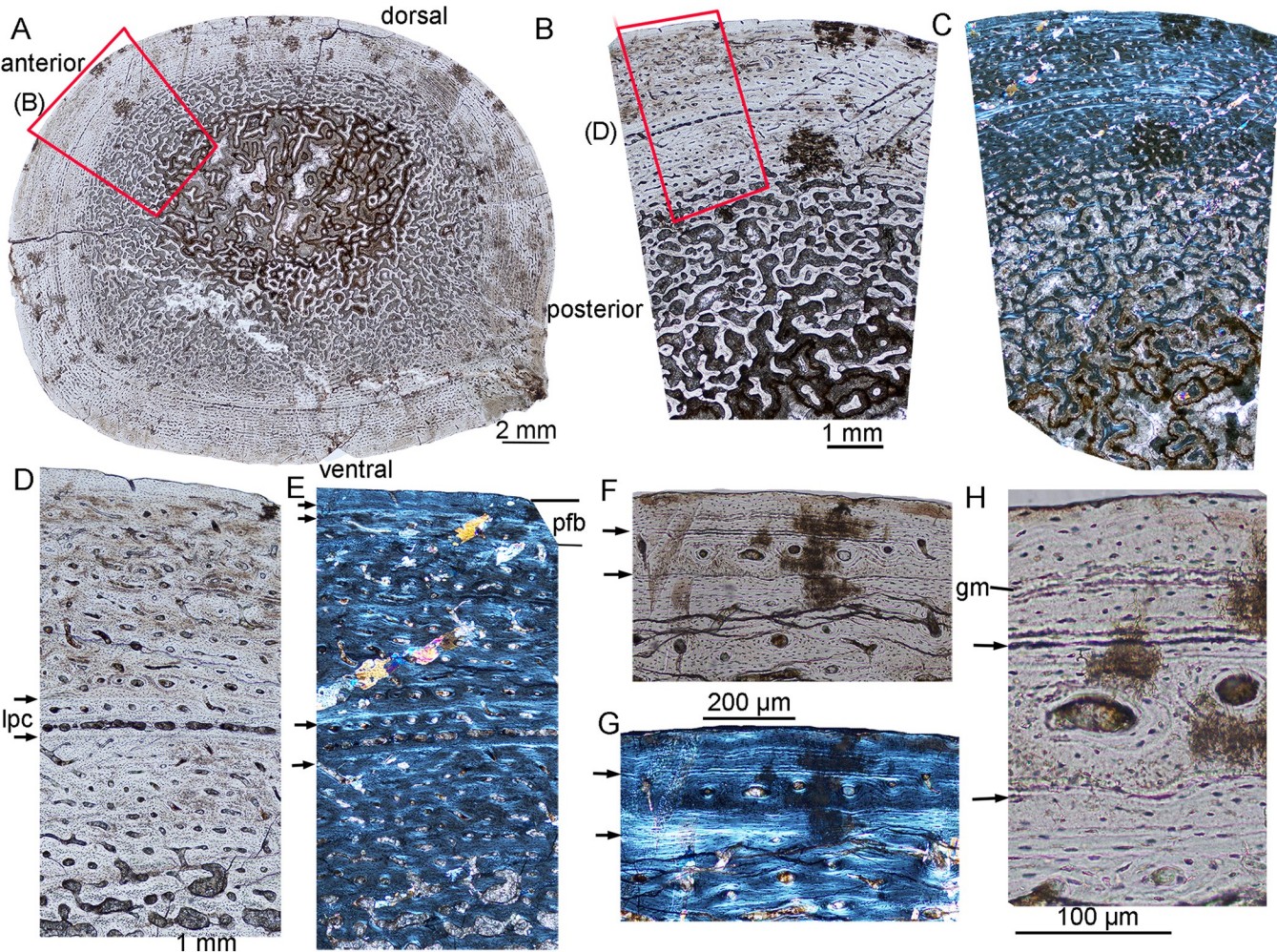

**Fig 5. Bone microstructure of the femur in late subadult *Lystrosaurus* IVPP V26547. A**, mid diaphyseal cross-section. **B**, partial cross-section under normal light. **C**, the same cortex as (B) under crossed plane-polarized light. **D**, enlargement of (B) under normal light, showing LAGs and parallel-fibered bone in the outer cortex. **E**. the same cortex as (D) under polarized light, showing LAGs and parallel-fibered bone. **F**, the outermost cortex under normal light showing LAGs and vascular canals; **G**, the same region as (F) showing parallel-fibered bone; **H**, enlargement of (F) showing LAGs, osteocyte lacunae and primary osteons. The arrows denote LAGs. **Abbreviation**: **gm,** growth marks; **lpc,** large primary canals.

## Early subadult

The early subadult stage was represented by IVPP V26545. The cross-sections of the fibula, radius, and rib of IVPP V26545 were sampled for observation (Fig 4). The cross-section of the fibula is subtriangular in outline with a diameter of 13.0 (AP) × 9.8 (ML) mm. The cortex consists of fibrolamellar bone. Most of the vascular canals are longitudinally arranged circumferentially around the bone, forming primary osteons. They are abundant in the inner region with a cortical porosity of about 35% (measured in the lateral cortex), but the number and size decrease from the mid-cortex (cortical porosity about 22%) to the periphery (cortical porosity about 10%). Secondary osteons are restricted to the inner cortex (Fig 4B), which consists of compact coarse cancellous bone (CCCB). A distinct LAG is present near the mid-region of the cortex (Fig 4A), and it separates into two closely spaced rest lines in some regions (Fig 4B). The medullary region is filled with trabeculae (Fig 4A).

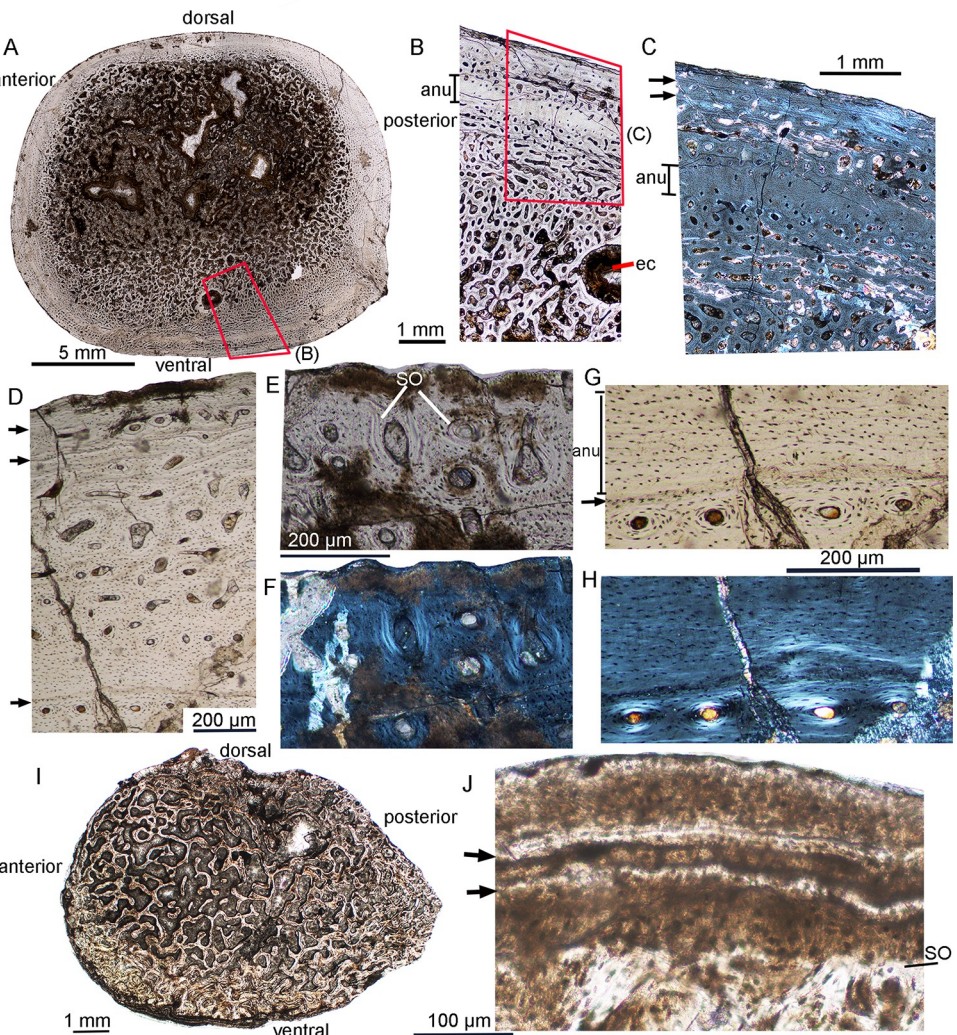

**Fig 6. Bone microstructure in late subadult *Lystrosaurus* IVPP V26546 and IVPP V26548.** (A-H) IVPP V26546. **A**, whole cross-section; **B**, partial transverse section of the femur under normal light; **C**, the outer cortex under polarized light showing parallel-fibered bone and LAGs near the periphery; **D**, enlargement of the outer cortex showing LAGs; **E**, the outermost cortex showing secondary osteons under normal light; **F**, the same region as (E) under polarized light; **G**, enlargement of the annulus showing the organized osteocyte lacunae; **H**, the same region as (G) under polarized light; (**I-J**) IVPP V26548. **I**, bone microstructure of the dorsal rib under normal light; **J**, enlargement of the outer region under normal light showing the LAGs. The arrows denote LAGs. **Abbreviation**: **anu**, annulus; **ec**, endosteal cavity; **so**, secondary osteons.

The radius is more robust than the fibula, with a diameter of 17.5 (ML) × 14.9 (AP) mm. The cortex mainly consists of woven fibered bone. The vascular canals are primarily longitudinal but there are also some irregular canals in the mid-cortex (Fig 4C). Secondary osteons and erosional cavities are present in the inner cortex. A clear LAG, as well as a thin layer of parallel-fibered bone, are present at the mid-region as in the fibula (Fig 4D and 4E). Periosteally from the LAG the density of vascular canals is gradually reduced but are still abundant (Fig 4C). The medullary cavity is large with a diameter of about one-third that of the whole cross-section.

The cross-section of the rib is teardrop-shaped, with a diameter of about 11.0 (AP) × 6.5 (ML) mm (Fig 4F and 4G). The compact bone is very thin (the average thickness is about 0.6

mm) due to a highly remodeled inner and mid-cortex. The medullary cavity is filled with trabeculae. The cortex consists of parallel-fibered bone tissues with only a few vascular canals (Fig 4G). The osteocyte lacunae are still abundant and disorganized. A double LAG is present in the outer cortex suggesting one year as the condition in the fibula and radius.

## Late subadult

The bone tissue of the femora of IVPP V26546 and IVPP V26547 shows late subadult tissues. In IVPP V26547, the midshaft of the femoral cortex is elliptical in outline with a diameter of 23.5 (AP) × 19.5 (DV) mm (Fig 5). There is no free medullary cavity. The cortex is composed of fibrolamellar bone throughout, and the outer cortex consists of parallel-fibered bone matrix with a thickness of about 1 mm (Fig 5E). Primary osteons are predominantly longitudinal and abundant throughout the cortex except for the region near the mid-cortex LAGs and the outer cortex. Secondary osteons are found in the inner and mid-cortex regions. Two prominent LAGs are present in the mid-cortex (Fig 5D and 5E). The distance between these two LAGs is narrow (about 0.5 mm), and a thin layer of large primary canals are present around the whole cross-section (Fig 5A and 5D). In the outermost cortex, the vascular canals are rare and nearly absent (Fig 5F). Two LAGs, as well as growth marks (weak growth lines), are present near the periphery (Fig 5F–5H). The osteocyte lacunae are flattened and organized along the growth lines (Fig 5H). The bone histology suggests that this individual grew very slowly before it died, but there is no external fundamental system (EFS), which is present in modern mammals, some dinosaurs, and some dicynodonts, and indicates skeletal maturity [31, 39, 44–46].

In IVPP V26546, the sampled section near the distal end of the femur is elliptical with a diameter of 26.5 (AP) × 23.0 (DV) mm. It has a thin compact bone (Fig 6). The cortex consists of fibrolamellar bone except for the periphery. The vascular canals are longitudinally-oriented and relatively small in the outer cortex compared to the mid and inner cortices (Fig 6B). Secondary osteons are common throughout the cortex and are also shown near the periphery (Fig 6E and 6F). A large erosional cavity surrounded by endosteal bone was present in the inner region of the ventral cortex (Fig 6B). A LAG with a narrow avascular layer of an annulus is present in the mid-cortex (Fig 6B, 6C, 6G and 6H), and there are only a few vascular canals around. The osteocyte lacunae are flattened and organized within the annulus (Fig 6G and 6H). At least two LAGs are present near the periphery (Fig 6C and 6D). The parallel-fibered bone matrix with LAGs and rare vascular canals suggests that it grew slowly before it died.

In the largest specimen IVPP V26548, only one dorsal rib was sampled (Fig 6I). The cortex is strongly remodeled in the inner and mid-region, and even in the outermost cortex in some regions. A thin layer of compact bone is shown in the outermost region. It is nearly avascular, with at least two LAGs near the periphery. However, these lines did not extend to the outermost region, and there are still some open vascular canals, suggesting this individual was still growing when it died (Fig 6J).

## Discussion

The bone tissue of *Lystrosaurus* consists predominantly of fibrolamellar bone with large channels, similar to those from Africa and India [22, 23]. Three growth stages of *Lystrosaurus* described here were identified: a fast-growing juvenile stage, a slower-growing early subadult, and a late subadult that grew more slowly. The fast growth rates may have helped them to adapt to the environment quickly [31]. The late juvenile was characterized by uninterrupted fibrolamellar bone tissue and the subadult primarily by fibrolamellar bone tissue interrupted by LAGs. The late subadult is characterized by the presence of parallel-fibered bone and LAGs in the outermost cortex. Bone tissue type in *Lystrosaurus* is not only related to the growth

stages but also differs between the various limb bones, including the humerus, femur, tibia, fibula and radius. The humerus, femur and tibia show a mainly laminar to plexiform vascular arrangement with a high cortical porosity in all stages, whereas the vascular canals of the fibula and radius are mostly longitudinally oriented and reduced in density, resulting in a low cortical porosity. In addition, our samples indicate that the humerus and tibia have higher cortical porosity than that of other limb bones, but the parallel-fibered bone with growth marks near the periphery appeared earlier in the humerus than in other elements (IVPP V26544).

Skeletal maturity is usually represented by the presence of an EFS [39], which has been identified in some dicynodonts, including *Placerias* [44] and *Kannemeyeria* [46]. To date, no typical EFS has been detected in *Lystrosaurus*. Ray et al. (2005) found parallel-fibered bone in the outer cortex of a dorsal rib and a thin layer of lamellar bone in a vertebra at the adult stage, but they did not find a typical EFS with closely spaced LAGs near the periphery. In our samples, the parallel-fibered bone in the outer cortex appeared in the dorsal rib and other elements, representing the subadult stage here (Fig 4). Although there is no typical EFS detected, the presence of more LAGs and parallel-fibered bone tissue near the periphery suggests that it grew slowly before it died.

Botha et al. [19] found that all Early Triassic *Lystrosaurus* from South Africa have no more than one observed LAG and no parallel-fibered bone in the outermost cortex, which suggests that most individuals did not reach skeletal maturity [19]. A new study on bone microstructure of South African *Lystrosaurus* provided more evidence that the two Early Triassic species were not fully grown. Only three of 12 individuals in *L. declivis* and three of 20 individuals in *L. murrayi* have one LAG in sampled specimens [24]. However, LAGs seem to be more commonly distributed in *Lystrosaurus* from the Early Triassic of China. Within the seven individuals, three have more than one LAG and one has one LAG. At least one LAG is present in humeri ranging from 12 to 15.6 cm and femora ranging from 12 to 19 cm in *Lystrosaurus* from China. This size variation is the most common in Triassic *Lystrosaurus* based on our observation. Comparison of the known humeral length with a LAG between Chinese and South African *Lystrosaurus* from the Early Triassic [24], showed that their size is not notably different (S4 Fig). However, Chinese *Lystrosaurus* tends to have more LAGs, which may be affected by different species or differing environmental conditions from South Africa, but this still needs study of more samples to approve.

The bone histology in the four *Lystrosaurus* species from the Lower Triassic of South Africa is structurally similar [31, 47] to the Chinese sample. However, IVPP V26545 has been classified as an early subadult and has only one LAG despite its humeral length being 15 cm. All the other individuals with humeral lengths larger than 12 cm (including estimated value) have more than two LAGs and are classified as late subadults (Table 1). In addition, the ratio of the humerus to ulna length is about 1.4, which is larger than that of IVPP RV35012 (1.2). This may indicate a different species of *Lystrosaurus*, which needs to be clarified in the future.

The presence of LAGs in *Lystrosaurus* from China suggests that many *Lystrosaurus* have reached at least the subadult stage, unlike Triassic *Lystrosaurus* from Africa where subadults are rare [19]. This difference may be caused by differing environmental conditions in these areas [23, 48]. A new study found evidence of prolonged and repeated metabolic stress in the tusk microstructure of *Lystrosaurus* from the Early Triassic of Antarctica that was thought to be extreme photoperiod seasonality [49]. During the Early Triassic, South Africa was located close to S60° , whereas the Juggar Basin lay near N35° [50, 51]. It indicates that the former region had stronger seasonality than the latter. Although evidence from calcic vertisols indicates that some regions of the Karoo Basin exhibited a wetting trend towards the *Daptocephalus-Lystrosaurus* Assemblage-Zone boundary [9], the hot and arid environment of the Karoo Basin in the Early Triassic has been widely accepted [6, 7, 47]. These harsh environmental

conditions may have resulted in high mortality rates of *Lystrosaurus*, with most individuals dying at an early stage [19]. But this hypophysis still needs more evidence in the future. By comparison, the climate is inferred to have been semiarid-subhumid in the Early Triassic of North China [35, 52, 53], which may indicate better environmental conditions than that in South Africa. However, the presence of more LAGs, especially double LAGs, may also suggest a harsh environmental condition in the Early Triassic of North China [48]. But the Chinese *Lystrosaurus* could still survive to a subadult stage probably by means of other survival strategy, such as a seasonal reduction in metabolic activity [49].

Another important issue is whether the body size of *Lystrosaurus* was reduced in the Early Triassic due to the harsh environment such as arid weather. Body size reduction has been shown in non-mammalian eutheriodont therapsids during the end-Permian mass extinction [11], and a reduction is also observed in *Lystrosaurus* from the Karoo Basin of South Africa, but this may be the result of the young stage of the body (indicated by rare LAGs) in the Early Triassic based on bone microstructure [19, 24]. Interestingly, the body size of *Lystrosaurus* from the lower section of the Guodikeng Formation (late Permian) such as *L. youngi* is smaller than that from the upper section of the Jiucaiyuan Formation, contrasting with the condition in South Africa [33]. This may indicate that the known material of *L. youngi* represents immature individuals, whereas later Chinese *Lystrosaurus* from the Early Triassic are at least subadult. However, the bone microstructure of *Lystrosaurus* from the Guodikeng Formation is poorly preserved, and more material needs to be assessed in the future to confirm this hypothesis.

The lifestyle of *Lystrosaurus* has been argued to be aquatic/semiaquatic [22], fully terrestrial, or burrowing [20, 21, 24]. In recent decades, this controversy has concentrated on aquatic or burrowing habits, which may have benefitted its survival in the hot environment of the Early Triassic. The aquatic/semiaquatic lifestyle was supported by the bone microstructure showing a thick cortex and porous vascular canals [22], and linear discriminant analysis based on the compactness profile and body size parameters in extant animals [54, 55]. The available cortical thicknesses of samples described here are between 19% - 41%, which is similar to *Lystrosaurus* from South Africa [22]. In sum, our results also show that *Lystrosaurus* has a thick cortex as in extant aquatic animals, such as *Hippopotamus* [56]. However, the thick cortex is not only shown in aquatic animals but also reported in extant burrowing animals and nonmammalian therapsids [43]. Therefore, a thick cortex could not provide confident evidence of an aquatic or semiaquatic lifestyle for *Lystrosaurus*. Moreover, juvenile *Lystrosaurus* have been found inside burrow casts indicating that they were active burrowers [25, 26].

Whether *Lystrosaurus* had a substantial aquatic component to its lifestyle in addition to burrowing could be elucidated by analyzing stable isotopes in the future. Previous studies suggest that aquatic and semi-aquatic vertebrates have $\delta^{18}O$ values significantly lower than those of coexisting terrestrial animals [57–59]. Oxygen and calcium isotopes all indicated that spinosaurids were aquatic or semiaquatic like modern crocodiles [60, 61], and they could also provide more evidence for the lifestyle of *Lystrosaurus*. Furthermore, $\delta^{18}O$ isotopes have already been used in some late Permian dicynodonts, and the result shows that *Endothiodon* may have had a semi-aquatic lifestyle, whereas *Tropidostoma* was more terrestrial [62].

## Conclusion

Here we have studied the bone histology of *Lystrosaurus* from the Early Triassic of China based on thin sections of seven individuals. The results indicate that *Lystrosaurus* grew rapidly in the early stages as previously shown in *Lystrosaurus* from South Africa and India. However, the presence of LAGs in the largest individual suggests that the growth of *Lystrosaurus* was

interrupted in the late stage. LAGs are rare in Triassic *Lystrosaurus* from South Africa but they are more abundant in the *Lystrosaurus* from Xinjiang, China. We propose that most *Lystrosaurus* from Xinjiang had reached the subadult stage, whereas those from South Africa died during their juvenile stage. This distinction may be caused by different environmental conditions, but more samples from the Early Triassic of China are needed to test this hypothesis in the future.

## Supporting information

**S1 Fig. Sampled positions of all specimens in this study. A**, IVPP V26543, arrow denotes the sampled position of the right tibia in lateral view; (**B, C**) IVPP V26544. **B**, the sampled positions of the right tibia and fibula in anterior view; **C**, the sampled position of the left humerus in ventral view; **D**, IVPP V26542, the distal end of the femur in ventral view; (**E-G**) IVPP V26545. **E**, the sampled position of the left radius (and ulna) in anterior view; **F**, the sampled position of the fibula; **G**, the sampled position of the rib; **H**, IVPP V26547, arrow denotes the sampled position of the femur in dorsal view; **I**, IVPP V26546, arrow denotes the sampled position of the femur in ventral view; **J**, IVPP V26548, arrow denotes the sampled position of the rib.
(PDF)

**S2 Fig. Positions for measurements of the cortical thickness in avialble thin sections. A**, IVPP V26543, tibia; **B**, IVPP V26544 tibia; **C**, IVPP V26544, humerus; **D**, IVPP V26544, fibula; **E**, IVPP V26542, femur; **F**, IVPP V26545, fibula; **G**, IVPP V26545, radius; **H**, IVPP V26546, femur; **I**, IVPP V26547, femur. The yellow lines denote measurements of the cortical thickness. The red lines denote measurements of cross-sectional diameters, and the green circles denote outlines of the medullary cavities.
(PDF)

**S3 Fig. Bone microstructure and the sampled areas for calculating cortical porosity in all thin sections.**
(PDF)

**S4 Fig. Comparison of the humeral length between Chinese and South Africa *Lystrosaurus*.**
(PDF)

**S1 Table. Raw measuremental data for caculating cortical thickness and porosity.**
(XLSX)

## Acknowledgments

We thank Zhang Shukang and Wu Rui for helping to make bone sections, Tong Jinnan and Chu Daoliang for their helpful discussion. We thank Jennifer Botha, Megan Whitney, and Zoe Kulik for their comments on a previous edition of this paper. Megan Whitney and Jennifer Botha also reviewed this version and have provided more constructive and useful comments. We thank the editor Jörg Fröbisch, and the reviewer Christian F. Kammerer and two other anonymous reviewers for their very helpful comments on this manuscript.

## Author Contributions

**Resources:** Jun Liu.

**Writing – original draft:** Fenglu Han.

**Writing – review & editing:** Qi Zhao, Jun Liu.

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
