## [Decision Letter · Decision Letter 0]

24 Sep 2020

PONE-D-20-17684

Bone histology of Lystrosaurus (Therapsida: Dicynodontia) from the Lower Triassic of North China, and its implication for lifestyle and environments after the end-Permian extinction

PLOS ONE

Dear Dr. Han,

Thank you for submitting your manuscript to PLOS ONE. After careful consideration, we feel that it has merit but does not fully meet PLOS ONE’s publication criteria as it currently stands. Therefore, we invite you to submit a revised version of the manuscript that addresses the points raised during the review process.

We look forward to receiving your revised manuscript.

Kind regards,

Jörg Fröbisch, Ph.D.

Academic Editor

PLOS ONE

Journal Requirements:

2. Please upload a copy of Supporting Information Figure S1 which you refer to in your text on page 4.

Additional Editor Comments (if provided):

Dear authors,

apologies that the decision process took so long. It was extremely difficult to receive reviews covering the taxonomic expertise on the one hand and more importantly also the histological expertise on the other hand. Please pay detailed attention to the comments made by ALL reviewers but specifically to the comments and criticism of Reviewer 4. For this manuscript to be acceptable for publication in Plos One, it is essential that you provide additional comparative data with regards to South African Lystrosaurus specimens as well as reconsider the interpretation of your own data, specifically with respect to the evidence for an external fundamental system (EFS), which is rather unconvincing based on the currently provided data.

If you feel that you can address all of the points raised by the reviewers and by myself, I'm looking forward to a revised version of your manuscript.

Best regards,

Jörg Fröbisch

Reviewers' comments:

Reviewer's Responses to Questions

**Comments to the Author**

1. Is the manuscript technically sound, and do the data support the conclusions?

Reviewer #1: Yes

Reviewer #2: No

Reviewer #3: Yes

Reviewer #4: Partly

2. Has the statistical analysis been performed appropriately and rigorously? 

Reviewer #1: Yes

Reviewer #2: N/A

Reviewer #3: Yes

Reviewer #4: N/A

3. Have the authors made all data underlying the findings in their manuscript fully available?

Reviewer #1: Yes

Reviewer #2: No

Reviewer #3: Yes

Reviewer #4: No

4. Is the manuscript presented in an intelligible fashion and written in standard English?

Reviewer #1: No

Reviewer #2: Yes

Reviewer #3: Yes

Reviewer #4: Yes

5. Review Comments to the Author

Reviewer #1: General comments: This manuscript describes the bone histology of a growth series of Lystrosaurus specimens from the Early Triassic of western China. Lystrosaurus is a famous survivor of the Permo-Triassic mass extinction, and work on specimens from South Africa suggests that there was a shift in growth and life history patterns in the genus going across the Permo-Triassic boundary. This change has been hypothesized to have contributed to the survival of Lystrosaurus and its success in the Early Triassic. In this context, the bone histology, growth, and life history patterns of specimens from China is of definite interests because of the insight they can provide about whether the factors that were important for Lystrosaurus survival in Africa were also at work in other geographic areas. The authors provide a fairly detailed and well-illustrated description of the bone histology of their specimens,and they find that the Chinese specimens are consistent with a more extended life history in the Early Triassic. The paper is novel in providing the first description of Chinese Lystrosaurus bone histology, and I think it will be an important reference that will be used in the ongoing discussion of the causes and effects of the Permo-Triassic extinction in the terrestrial realm. The discussion section is fairly short, but I think the authors do a good job of not over-stepping the limits of what can be said from their (admittedly fairly small) sample, and ultimately the primary value of the paper is in the description of the histology. Overall, I think the paper can proceed with minor revisions. That said, I recommend having someone who really focuses on histological work take a look at the paper as well. The description seems accurate to me given what is in the figures, but there may be subtleties that I missed because histology is not my primary focus. The language is good overall, but I noted some areas where minor revisions are needed for usage and clarity.

abstract line 4: remove ‘the’ before growth

abstract line 6: move ‘for the first time’ to after China

abstract line 8: change suggest to indicate

abstract line 9: change line to lines

abstract line 12: remove ‘the’ before growth

introduction line 4: It might be good to note that there is some debate about this, at least for the extinction in the Karoo Basin, such as gGastaldo et al. (2020; Journal of Sedimentary Research) and references cited therein.

page 3 line 2: instead of curve, to would recommend using downturned, since the shape is more angular than curved in most Lystrosaurus species.

page 3 line 3: The report of Lystrosaurus in Zambia is incorrect (See Angielczyk et al. 2014; Early Evolutionary History of the Synapsida book), so I recommend not citing the King and Jenkins paper. Frobisch (2009; Earth-Science Reviews) would be a more comprehensive paper to cite here (although, it still includes the incorrect Zambian record).

page 3 line 4: change their to its

page 3 line 6: It might be worth noting that the semi-aquatic ecology of Lystrosaurus has been disputed including using bone histology data (e.g., King 1991; King and Cluver 1991, both Historical Biology; Botha-Brink and Angielczyk 2010, ZJLS).

page 3 line 7: change borrow to burrow; also remove ‘from the’ later in the sentence.

page 3, paragraph 2: A couple comments here. 1) remove ‘the’ before minaturization. 2) I suggest a little re-phrasing of the results from the Botha-Brink et al. (2016) paper. That paper did indicate that LAGs were more common in Permian specimens, which was interpreted as evidence that the Permian species had a more extended ontogeny that extended over several years. This contrasts with the prevalence of small individuals with few or no LAGs in the Triassic species, which was part of the evidence suggesting breeding young (the other part of the evidence were the demographic simulations that suggested that breeding young could be an effective survival strategy when life expectancy was low).

page 3, last paragraph, line 2: remove ‘the’ before Lystrosaurus

material and methods line 6: I’m not sure if the specimens you sectioned had associated skulls. If they do not, I recommend saying that ID to species level is currently not possible because Chinese Lystrosaurus taxonomy (and Lystrosaurus taxonomy in general) is currently based on skull characters. If they have skulls, you might say that you didn’t try to ID them to species level given the uncertainties about Chinese Lystrosaurus species-level taxonomy that you mentioned above.

material and methods line 8: Morphology refers to macroscopic features, whereas histology refers to microscopic tissue structure. So you should say: “...South Africa is quite similar, although more LAGs...” You should also make this change in the next sentence when you make your predictions about the Chinese species.

material and methods line 10: Identical histology might be too strong. Similar is probably a better prediction, assuming there aren’t differences due to environmental conditions (as you note).

Materials and methods, paragraph 2, line 3: complete instead of completed; also at the end of this sentence, change bones to bone. It would be good to note whether and how the specimens were documented before sectioning (e.g., e.g., if photographs or measurments were taken, if molds/casts were made, etc.).

Page 5, line 1: ground instead of grinded

Page 5, line 3: remove ‘the’ before transmitted

Table 1: How did you assign your specimens to growth stages? Is this something that was done a priori, and if so what was it based on (Size? Degree of ossification and development of muscle scars and articular surfaces? Something else?). You should include this information in the methods section.

Page 6, paragraph 1: This paragraph uses a mixture of present tense and past tense. Please revise it so it consistently uses past tense.

Page 6, line 2-3: You should state why it was difficult to measure cortical porosity for a whole cross section. Was it because of breakage, differential preservation of the bone tissue in different areas, something else? Also, It might be good to elaborate a bit on how you chose areas to measure when that was necessary, maybe the average number of areas you needed to use (in addition to the maximum you report), and whether using different numbers of areas had any effect on the results.

Fig. 1: I recommend reorganizing this figure slightly so that the overview of the specimen showing the focal areas is in panel A (similar to Fig. 2).

juvenile line 2: change to: All the corticiesare composed of a woven-fibered bone matrix, with a…

juvenile, line 5: when you mention cortical thickness and porosity values here and elsewhere in the text, you should call out Table 1. It would also be good if you can be more consistent in mentioning those values for the different elements in the text.

Fig. 2 and Fig. 2 caption: H and I should be switched in the figure in the caption given that you discuss what is currently I (but should be H) first. Also, panel O is a little dark. Can you adjust the brightness a bit on the image?

page 12, first full paragraph line 5: remove ‘the’ before parallel-fibered

Fig. 3: Please add boxes to the overview image in panel A to show where the higher magnification images are located.

Page 13, line 3: I think you mean: “The number and diameters of the vascular canals in the narrow region...”

page 13, line 7: change trabecular to trabeculae

Fig. 5 : Please add boxes to the overview image in panel A to show where the higher magnification images are located.

Page 16, line 2: An EFS was reported in the dicynodont Placerias by Green et al. (2010; Palaeontology).

Page 16, IVPP V26546, line 1: change to: “It has thin compact...”

Fig. 6 : Please add boxes to the overview image in panel A to show where the higher magnification images are located.

Page 16, last line: change suggests to suggest

discussion line 1: change tissues to tissue

page 18, line 9: change most to mostly

page 18, line 10: change than to that

page 18, first full paragraph, line 1: Change to “Ray et al. (2005) found parallel-fibered...”

page 18, first full paragraph, line 5: change to: “...although there is not typical...”

page 18, first full paragraph, line 7: Bone histology has been looked in a a pretty wide range of dicynodonts now, and an EFS has only been observed in a couple of instances. Therefore, it seems odd to suggest that it may be present in Lystrosaurus, especially when you didn’t find any evidence of one in any of your specimens.

Page 19, line 9: As noted above, some research has suggested wetter conditions in the Karoo at this time. It would be good to note that possibility here as well.

Page 19, second paragraph line 1: change to: “...was reduced in the Early Triassic...”

Page 19, last line: It might be good to qualify this a bit by stating that most collected Early Triassic specimens are at least subadults. My sense is that Lystrosaurus hasn’t been documented in a comprehensive enough way to have an unbiased sense of the relative abundances of different size classes.

Page 20, first full paragraph: The other thing that’s important to note here is that the bone microstructure of Lystrosaurus doesn’t seem to differ all that much from other dicynodonts (e.g., Botha-Brink and Angielczyk 2010). So either that would imply that most dicynodonts were semi-aquatic or that Lystrosaurus was terrestrial like other dicynodonts (with dicynodonts in general showing some fairly consistent differences in microstructure to modern terrestrial tetrapods).

Page 20, first full paragraph, next to last sentence: It would be good to cite Rey et al. (2020; Palaeo3) here since it is a study that uses stable isotope data to make inferences about habitat preferences of a diynodont.

Reviewer #2: There are some critical flaws that I think could only be addressed with an adequate comparative dataset of Permian Lystrosaurus specimens, and it's unclear to me whether the research team has adequately surveyed Permo-Triassic therapsid histology at all. Based on the abundant material of Permo-Triassic Lystrosaurus that I've thin-sectioned and seen firsthand, the South African material is nearly identical to the Chinese specimens. This study is hindered by a lack of any comparative specimens, and would further benefit from a more quantitative scientific approach.

Reviewer #3: The frequent presence of LAGs in the Chinese specimens is interesting and well worth publishing. I have made a number of copy edits and comments on some unclear parts of the manuscript in the attached file, which should be addressed before this contribution can be accepted.

Reviewer #4: This study describes the bone histology of seven Early Triassic Lystrosaurs from Xinjiang, China. The authors describe an ontogenetic series of rapid growth, with some periods of interrupted growth, and eventually a slowed, adult-stage growth. The authors compare these findings to those from the bone histology of South African Lystrosaurus and detail that although signals of arrested growth are rarely reported from South Africa, they were found in their sample. This suggests that perhaps the samples from South Africa are still in juvenile/subadult stage. The authors suggest that this finding could be explained by environmental conditions that are thought to have been more extreme in South Africa than in China during the Early Triassic resulting in a greater die off of individuals before skeletal maturity in South Africa. The authors also discuss their findings as it relates to the lifestyle of Lystrosaurus, refuting pervious hypotheses that the genus was aquatic/semiaquatic.

I have reviewed a previous version of this manuscript and found the grammar and organization to be much improved. There were still some line edits of which I have added to my marked version of the manuscript. I have also added comments that are both specific and general to this document, but in addition I have a few general comments about the manuscript. I think the figures are incredibly well presented and it is important this data is published. However, I think there are very major revisions in the discussion points/conclusions as well as some methodology that requires attention prior to publication.

Generally, I think the authors need to make a more compelling argument that their sample represents an ontogenetic series, independent of the histology. Table 1 I believe is supposed to address this, but the organization is difficult for the reader to follow, especially when I can’t compare a single element across the sample. To address this I think the authors, using more complete specimens and previously established ratios between elements, could come up with estimates of a single element for each individual in their sample. Alternatively, a single figure that images all of the samples at the same scale would help the reader confirm the authors’ designations of ontogenetic stages. I believe there is a supplementary figure, however this figure was not included in my review document so I am unable to say if it would help to understand the relative sizes of these individuals. This is the only reason why I answered that not all data was available. In such a figure, I would also include images of the largest South African and Indian Lystrosaurus elements so that the reader can see just how much larger the Chinese specimens are.

The methodology still requires significant revision/clarity. I would review previous histological papers that measure cortical thickness. I think for cortical thickness, presenting a range is more important that an average, especially in some specimens that are asymmetrically arranged. Cortical porosity also needs to be addressed. The description as it stands does not seem systematic and completely subjective. Subsampling vascularity is ok, but the locations (medially, laterally, posterior, anterior, endosteally, mid-cortex, periosteally) need to be systematically sampled similarly across the entire sample. Without this, it is difficult to review the values reported.

In the descriptions differences in cortical porosity need to be explicitly stated. There are many instances where the percentage for one region (e.g. endosteal) of the cortex is reported and then the authors state there is a reduction in vascularity towards the periosteal surface, but no percentage is reported. Consistency is important in interpreting the descriptions.

It is important that the authors describe the anatomical context for the images (i.e. medial, lateral sides etc.). This can be done in the image itself (e.g. compass-style arrows) or in the figure caption.

Finally, I do not think the data here is sufficient to refute the hypothesis that Lystrosaurus was aquatic. I agree entirely that it is unlikely Lystrosaurus was not aquatic, however, I don’t believe the data as currently presented in this manuscript supports this. In fact, the thick cortical walls appear very similar to aquatic taxa, which is why previous studies have suggested an aquatic lifestyle. However, there are studies that demonstrate this is not always a reliable proxy for lifestyle (e.g. Houssaye et al. 2016). I would recommend one of two courses of action. Either remove the discussion of the aquatic lifestyle from this manuscript OR discuss how cortical thickness does not always accurately predict habitat.

6. PLOS authors have the option to publish the peer review history of their article (what does this mean?). If published, this will include your full peer review and any attached files.

Reviewer #1: No

Reviewer #2: No

Reviewer #3: **Yes: **Christian F. Kammerer

Reviewer #4: **Yes: **Megan Whitney

---

## [Author Response · Author response to Decision Letter 0]

26 Nov 2020

Dear Editor and Reviewers:

Thank you for your letter and for the reviewers’ comments concerning our manuscript entitled “Bone histology of Lystrosaurus (Therapsida: Dicynodontia) from the Lower Triassic of North China, and its implication for lifestyle and environments after the end-Permian extinction” (ID: PONE-D-20-17684). Those comments are all valuable and very helpful for revising and improving our paper, as well as important for highlighting the significance of our work. We have studied the comments carefully and made a number of corrections to address them, which we hope to meet with approval. The revised portions of the MS were marked by red colors. The main corrections to the paper and the responses to the editor and reviewer’s comments are as following:

Editor comments:

Please pay detailed attention to the comments made by ALL reviewers but specifically to the comments and criticism of Reviewer 4. For this manuscript to be acceptable for publication in Plos One, it is essential that you provide additional comparative data with regards to South African Lystrosaurus specimens as well as reconsider the interpretation of your own data, specifically with respect to the evidence for an external fundamental system (EFS), which is rather unconvincing based on the currently provided data.

REPLY: We have modified our MS carefully according to all reviewers’ comments. We have also added more comparisons between Chinese Lystrosaurus and South African Lystrosaurus, including both bone microstructure and body size. In addition, We have deleted the statement of the possible presence of EFS in Lystrosaurus based on current evidence, and we just mentioned that “there is still no typical EFS found in Lystrosaurus”.

Reviewer1 

introduction line 4: It might be good to note that there is some debate about this, at least for the extinction in the Karoo Basin, such as Gastaldo et al. (2020; Journal of Sedimentary Research) and references cited therein

REPLY: We have added the debate and related references in the MS. 

page 3 line 2: instead of curve, to would recommend using downturned, since the shape is more angular than curved in most Lystrosaurus species.

REPLY: Corrected.

page 3 line 3: The report of Lystrosaurus in Zambia is incorrect (See Angielczyk et al. 2014; Early Evolutionary History of the Synapsida book), so I recommend not citing the King and Jenkins paper. Frobisch (2009; Earth-Science Reviews) would be a more comprehensive paper to cite here (although, it still includes the incorrect Zambian record).

REPLY: We have changed the citation as suggested.

page 3 line 6: It might be worth noting that the semi-aquatic ecology of Lystrosaurus has been disputed including using bone histology data (e.g., King 1991; King and Cluver 1991, both Historical Biology; Botha-Brink and Angielczyk 2010, ZJLS).

REPLY: We have added more citations here.

page 3, paragraph 2: A couple comments here. 1) remove ‘the’ before minaturization. 2) I suggest a little re-phrasing of the results from the Botha-Brink et al. (2016) paper. That paper did indicate that LAGs were more common in Permian specimens, which was interpreted as evidence that the Permian species had a more extended ontogeny that extended over several years. This contrasts with the prevalence of small individuals with few or no LAGs in the Triassic species, which was part of the evidence suggesting breeding young (the other part of the evidence were the demographic simulations that suggested that breeding young could be an effective survival strategy when life expectancy was low).

REPLY: Modified as suggested.

material and methods line 6: I’m not sure if the specimens you sectioned had associated skulls. If they do not, I recommend saying that ID to species level is currently not possible because Chinese Lystrosaurus taxonomy (and Lystrosaurus taxonomy in general) is currently based on skull characters. If they have skulls, you might say that you didn’t try to ID them to species level given the uncertainties about Chinese Lystrosaurus species-level taxonomy that you mentioned above.

REPLY: Thank you for your suggestion. Our specimens only include postcranial materials. Modified as suggested.

material and methods line 8: Morphology refers to macroscopic features, whereas histology refers to microscopic tissue structure. So you should say: “...South Africa is quite similar, although more LAGs...” You should also make this change in the next sentence when you make your predictions about the Chinese species.

REPLY: Corrected. Furthermore, we moved this part to the discussion as suggested by another reviewer.

material and methods line 10: Identical histology might be too strong. Similar is probably a better prediction, assuming there aren’t differences due to environmental conditions (as you note).

REPLY: Modified as suggested.

Materials and methods, paragraph 2, line 3: complete instead of completed; also at the end of this sentence, change bones to bone. It would be good to note whether and how the specimens were documented before sectioning (e.g., e.g., if photographs or measurments were taken, if molds/casts were made, etc.).

REPLY: We have added this information in paragraph 2. 

Table 1: How did you assign your specimens to growth stages? Is this something that was done a priori, and if so what was it based on (Size? Degree of ossification and development of muscle scars and articular surfaces? Something else?). You should include this information in the methods section.

REPLY: Growth stages were identified based on bone histological information, and we have added this information in the footnotes of Table 1.

Page 6, paragraph 1: This paragraph uses a mixture of present tense and past tense. Please revise it so it consistently uses past tense.

REPLY: Corrected. 

Page 6, line 2-3: You should state why it was difficult to measure cortical porosity for a whole cross section. Was it because of breakage, differential preservation of the bone tissue in different areas, something else? Also, It might be good to elaborate a bit on how you chose areas to measure when that was necessary, maybe the average number of areas you needed to use (in addition to the maximum you report), and whether using different numbers of areas had any effect on the results.

REPLY: We have added this information and modified the sentence as “It was difficult to obtain the cortical porosity of a whole cross-section for each specimen because of partially breakage or poor preservation in some areas of these sections.”

 Cortical porosity is usually reduced from the inner region to outside, but they are usually (not always) similar on all sides (medial, lateral, anterior, posterior). We subsampled ten areas from the inner to the outermost cortex in all sections (except IVPP V26546 was selected 12 areas) to calculate average vascular porosity. The sampled areas were chosen from the inner region to the periphery on different parts of the whole cross-sections, so they could reflect the value of cortical porosity. However, some bone sections show unclear bone histology in some regions, which may affect the results. Cortical porosities of those poorly preserved bone sections were labeled to be “estimated values”. We have provided the figures showing the sampled area (S3 Fig).

Fig. 1: I recommend reorganizing this figure slightly so that the overview of the specimen showing the focal areas is in panel A (similar to Fig. 2).

REPLY: Modified as suggested. 

juvenile line 2: change to: All the corticies are composed of a woven-fibered bone matrix, with a…

REPLY: Corrected. We have modified this sentence as “All the cortices are composed of fibrolamellar tissue, with a high number of vascular canals, abundant globular osteocyte lacunae, and no LAGs, suggesting a rapid, constant growth rate.”

juvenile, line 5: when you mention cortical thickness and porosity values here and elsewhere in the text, you should call out Table 1. It would also be good if you can be more consistent in mentioning those values for the different elements in the text.

REPLY: We have added this information throughout the manuscript. 

Fig. 2 and Fig. 2 caption: H and I should be switched in the figure in the caption given that you discuss what is currently I (but should be H) first. Also, panel O is a little dark. Can you adjust the brightness a bit on the image?

REPLY: Modified as suggested. 

Fig. 3: Please add boxes to the overview image in panel A to show where the higher magnification images are located.

REPLY: Added.

Page 13, line 3: I think you mean: “The number and diameters of the vascular canals in the narrow region...”

REPLY: Thank you for pointing out. Corrected.

Fig. 5 : Please add boxes to the overview image in panel A to show where the higher magnification images are located.

REPLY: Added.

Page 16, line 2: An EFS was reported in the dicynodont Placerias by Green et al. (2010; Palaeontology).

REPLY: We have added more references here. 

Fig. 6 : Please add boxes to the overview image in panel A to show where the higher magnification images are located.

REPLY: We have added a box in panel A, showing the position of B, and also a box in panel B, showing the position of C. 

page 18, first full paragraph, line 7: Bone histology has been looked in a pretty wide range of dicynodonts now, and an EFS has only been observed in a couple of instances. Therefore, it seems odd to suggest that it may be present in Lystrosaurus, especially when you didn’t find any evidence of one in any of your specimens.

REPLY: We have modified this paragraph and conclusion here. The modified statement is “There is still no typical EFS detected in Lystrosaurus.”

Page 19, line 9: As noted above, some research has suggested wetter conditions in the Karoo at this time. It would be good to note that possibility here as well.

REPLY: we have added new references on wetter conditions.

Page 19, last line: It might be good to qualify this a bit by stating that most collected Early Triassic specimens are at least subadults. My sense is that Lystrosaurus hasn’t been documented in a comprehensive enough way to have an unbiased sense of the relative abundances of different size classes.

REPLY: Modified as suggested.

Page 20, first full paragraph: The other thing that’s important to note here is that the bone microstructure of Lystrosaurus doesn’t seem to differ all that much from other dicynodonts (e.g., Botha-Brink and Angielczyk 2010). So either that would imply that most dicynodonts were semi-aquatic or that Lystrosaurus was terrestrial like other dicynodonts (with dicynodonts in general showing some fairly consistent differences in microstructure to modern terrestrial tetrapods).

REPLY: Our samples support Lystrosaurus has a thick cortex, as in Lystrosaurus of Africa. But thick cortex is not only seen in aquatic animals but also shown in extant burrowing animals. Therefore, thick cortex could not provide confident evidence of the aquatic or semiaquatic lifestyle of Lystrosaurus, so we did not discuss the lifestyle of other dicynodonts. We have provided more discussion on Lystrosaurus lifestyle and how to solve this problem. 

Page 20, first full paragraph, next to last sentence: It would be good to cite Rey et al. (2020; Palaeo3) here since it is a study that uses stable isotope data to make inferences about habitat preferences of a diynodont.

REPLY: Thank you for the information. Cited.

Reviewer 2.

There are some critical flaws that I think could only be addressed with an adequate comparative dataset of Permian Lystrosaurus specimens, and it's unclear to me whether the research team has adequately surveyed Permo-Triassic therapsid histology at all. Based on the abundant material of Permo-Triassic Lystrosaurus that I've thin-sectioned and seen firsthand, the South African material is nearly identical to the Chinese specimens. This study is hindered by a lack of any comparative specimens, and would further benefit from a more quantitative scientific approach.

REPLY: Thank you for your suggestions. We have added more comparison to Lystrosaurus from South Africa on both bone histology and body sizes. We admit that our samples are limited, but they do show some interesting and clear features that unlike those of Lystrosaurus from South Africa, and we will provide more data on bone histology and other parts of Chinese Lystrosaurus in the future. 

Reviewer #3: The frequent presence of LAGs in the Chinese specimens is interesting and well worth publishing. I have made a number of copy edits and comments on some unclear parts of the manuscript in the attached file, which should be addressed before this contribution can be accepted.

REPLY: Thank you for your comments and detailed corrections for the grammar in the MS. We have checked the whole text and corrected all these things. Here are our replies to the main comments.

Page 4 Line 72. “Therefore, here we suppose that Lystrosaurus from the Taodonggou area of Xinjiang China should have identical morphology on bone histology even in different species. However, they may have distinct bone histological information from other places due to different environmental conditions”

Comment: This statement is unclear. Why should they have identical morphology?

REPLY: we have changed the word “identical” to “similar”. And moved this part to the discussion section. Here we mean that the previous work on Africa Lystrosaurus shows similar bone microstructure in different species, and Chinese Lystrosaurus may also have similar bone microstructure even in different species.

Page 6 Line 98. “Therefore, we subsampled areas (as many as ten) from the inner to the outermost cortex to calculate average vascular porosity.”

Comment: What is the minimum?

REPLY: We subsampled ten areas in all the sections. We have provided the sampled areas on photographs in the new supplementary files (S3 Fig). 

Page 11. Line 206 The number and diameter of the vascular canals in the narrow region is significantly reduced near the periphery.

COMMENT: This does not make sense to me.

REPLY: Thank you for pointing out. We have modified this sentence to “The number and diameter of the vascular canals are significantly reduced near the periphery.”

Page 12. Line 210 “Trabeculae? In the trabecular region?”

REPLY: Yes. We have corrected this word. 

Page 19 Recent papers by Kevin Rey on South African oxygen isotope work in dicynodonts would be worth citing and possibly discussing here.

REPLY: Thank you for your useful information. We have added this new reference here.

Reviewer #4: 

This study describes the bone histology of seven Early Triassic Lystrosaurus from Xinjiang, China. The authors describe an ontogenetic series of rapid growth, with some periods of interrupted growth, and eventually a slowed, adult-stage growth. The authors compare these findings to those from the bone histology of South African Lystrosaurus and detail that although signals of arrested growth are rarely reported from South Africa, they were found in their sample. This suggests that perhaps the samples from South Africa are still in the juvenile/subadult stage. The authors suggest that this finding could be explained by environmental conditions that are thought to have been more extreme in South Africa than in China during the Early Triassic resulting in a greater die-off of individuals before skeletal maturity in South Africa. The authors also discuss their findings as it relates to the lifestyle of Lystrosaurus, refuting previous hypotheses that the genus was aquatic/semiaquatic.

I have reviewed a previous version of this manuscript and found the grammar and organization to be much improved. There were still some line edits of which I have added to my marked version of the manuscript. I have also added comments that are both specific and general to this document, but in addition, I have a few general comments about the manuscript. I think the figures are incredibly well presented and it is important this data is published. However, I think there are very major revisions in the discussion points/conclusions as well as some methodology that requires attention prior to publication.

Generally, I think the authors need to make a more compelling argument that their sample represents an ontogenetic series, independent of the histology. Table 1 I believe is supposed to address this, but the organization is difficult for the reader to follow, especially when I can’t compare a single element across the sample. To address this I think the authors, using more complete specimens and previously established ratios between elements, could come up with estimates of a single element for each individual in their sample. Alternatively, a single figure that images all of the samples at the same scale would help the reader confirm the authors’ designations of ontogenetic stages. I believe there is a supplementary figure, however, this figure was not included in my review document so I am unable to say if it would help to understand the relative sizes of these individuals. This is the only reason why I answered that not all data was available. In such a figure, I would also include images of the largest South African and Indian Lystrosaurus elements so that the reader can see just how much larger the Chinese specimens are.

REPLY: Thank you for your suggestions. Sorry for unmodified this part last time. The postcranial skeleton of Lystrosaurus was not well studied, and we did not know the variation between different species and ontogeny, so we did not estimate values previously. We have rechecked more Lystrosaurus materials and provided estimated values based on element ratios of one complete skeleton (IVPP RV35012), please see the footnotes of Table 1 for detail information. 

In addition, sorry that we forgot to upload figure S1. In this figure, we have provided the accurate sampled positions of each element. 

The methodology still requires significant revision/clarity. I would review previous histological papers that measure cortical thickness. I think for cortical thickness, presenting a range is more important that an average, especially in some specimens that are asymmetrically arranged. Cortical porosity also needs to be addressed. The description as it stands does not seem systematic and completely subjective. Subsampling vascularity is ok, but the locations (medially, laterally, posterior, anterior, endosteally, mid-cortex, periosteally) need to be systematically sampled similarly across the entire sample. Without this, it is difficult to review the values reported.

REPLY: We have provided ranges for all cortical thickness and labeled the measurement positions in S2 Fig.

We have added more descriptions of how to measure cortical porosity. We choose ten areas that cover all the cortex. The sampled areas were from the inner region to the periphery on different parts of the whole cross-sections, and it indicates the average cortical porosity of the whole cross-section. Some bone sections show unclear bone histology in some regions, which may affect the results. Cortical porosities of those poorly preserved bone sections were labeled to be “estimated values”. We have also provided the figures showing the areas that we choose. Please see S3 Fig. 

In the descriptions differences in cortical porosity need to be explicitly stated. There are many instances where the percentage for one region (e.g. endosteal) of the cortex is reported and then the authors state there is a reduction in vascularity towards the periosteal surface, but no percentage is reported. Consistency is important in interpreting the descriptions.

REPLY: We have added more percentages to make it clear. 

It is important that the authors describe the anatomical context for the images (i.e. medial, lateral sides etc.). This can be done in the image itself (e.g. compass-style arrows) or in the figure caption.

REPLY: We checked all the specimens again and have added the directions for all the figures. 

Finally, I do not think the data here is sufficient to refute the hypothesis that Lystrosaurus was aquatic. I agree entirely that it is unlikely Lystrosaurus was not aquatic, however, I don’t believe the data as currently presented in this manuscript supports this. In fact, the thick cortical walls appear very similar to aquatic taxa, which is why previous studies have suggested an aquatic lifestyle. However, there are studies that demonstrate this is not always a reliable proxy for lifestyle (e.g. Houssaye et al. 2016). I would recommend one of two courses of action. Either remove the discussion of the aquatic lifestyle from this manuscript OR discuss how cortical thickness does not always accurately predict habitat.

REPLY: Sorry for the confusion. We did not refute the hypothesis that Lystrosaurus was aquatic. We prefer to keep this paragraph and have added more discussion here. Most of our samples support Lystrosaurus has a thick cortex, as in Lystrosaurus of Africa. But thick cortex is not only seen in aquatic animals but also shown in extant burrowing animals. Therefore, thick cortex could not provide confident evidence of the aquatic or semiaquatic lifestyle of Lystrosaurus. The lifestyle of Lystrosaurus is still controversial, but we provide more data for this issue. 

Other comments in the text

Page 4 Line 9. Comment: This is an important discussion point, but should not be in the materials and methods section. Perhaps move to the discussion when reviewing the variation.

REPLY: We have moved this paragraph to the discussion part.

Page 5 Line 5. Comment: This is a confusing sentence. I think to simplify and to set up the descriptions nicely just describe the histological criteria used to assign a growth line (especially annuli, since you use that term later) vs. LAG.

REPLY: We have modified this sentence as “LAGs are growth marked lines that can be traced around the full circumference. The annuli correspond to periods of slow growth by the presence of parallel-fibered bone and flattened osteons”. 

Page 5 Table 1. Comment: I think this table is really confusing. It's hard to compare the sizes between specimens since there are different elements. See overall notes for suggestions.

REPLY: We have added the estimated values as suggested.

Page 5 Table 1. Comment: If you are going to estimate values you have to describe how those estimations were made.

REPLY: We have added an explanation of how to estimate values in the footnotes. 

Page 7 line 7. Comment: This is very confusing and I'm not convinced it properly measures cortical thickness. Generally, I don't actually think the "diameter" measurements given in each of the description sections are properly stated either. I would suggest reporting mediolateral and anteroposterior widths (which is basically what I think the "diameter" measurements are?). I believe given the variability in cortical thickness around the bone, reporting the average is not helpful. Instead, maybe report a max and min %?

REPLY: We have modified the MS as suggested.

P10 L4 comment: Label these in the figure.

REPLY: Labeled as suggested.

P10L5 There are no secondary osteons. 

Comment: “It looks like there is in fact a massive one in 1C”

REPLY: we assigned it to be an enlarged resorption cavity rather than secondary osteons for its extremely large size. No Haversian system was presented around the erosion cavity.

P10 L6. This has already been described earlier in the paragraph

REPLY: we have deleted it. 

P10 L11.

I feel that wide-open is too casual of a term. Instead I would go with something like "enlarged canals". You could also just highlight the band of expanded vascularity with a bracket like: |-----|

REPLY: “enlarged canals” usually indicate secondary remodeling in our manuscript. But here it should be the primary bone. Therefore, we just described its shape here. We have added a bracket as suggested. 

P11 Line 16 The humerus is subtriangular in cross-section (midshaft diameter: 21×25 mm) and more robust than the tibia. 

Comment: I don't know what this means. In what way is it more robust? Greater cortical thickness?

REPLY: “subtriangular” means triangular but less angled. We have shown more figures in supplementary files (S3 Fig). “robust” means the humerus bone has a larger cross-section. Modified as suggested.

Page 12 Line 9 

Where within the cortex are these tissues found? How are they distributed throughout the cortex?

REPLY: We have modified this sentence as “The cortex mainly consists of fibrolamellar bone tissue, but parallel-fibered bone tissue is shown in the posterior cortex”.

Page 12 subadult line 3. “The medullary cavity is large (the diameter is about 5 mm)” comment: but not relative to the rest of the cross section...it may be large compared to the juvenile...but this is relative. It appears quite small to me in the figure.

REPLY: We have modified this sentence to “The medullary cavity is relatively large (the diameter is about 5 mm) compared to those of juvenile individuals”

P13. Line 2. What side of the bone is this on? It's a cool structure so some anatomical context would be appreciated.

REPLY: Added as suggested.

P14 Line 1. Also include mid-cortex and endosteal values. See general notes for more on this.

REPLY: We have added these values in the revised MS. 

P14 Fig. 4 Where are D and E from in C? See general notes for further comments on this.

REPLY: we have added a box in C to show the position of D and E.

P14. Line 16. The vascular canals are primarily longitudinal but there are also many irregular canals 

comments: Where? Label them in figure.

REPLY: We have added the labels in figure 4C.

P14 Line 18. Label these in C as well so there is context for D and E

REPLY: Added.

P15. Comments: The distinction between growth marks and LAGs should be clearly established in the methods section

REPLY: We have added their differences in the methods section. Growth marks and LAGs both form lines, but growth marks could not be traced on the whole circumference cortex bone. 

P16 figure 5. comment: Indicate from where in A. See general notes for more details

REPLY: We have added a box in A showing the position of B. 

P16 figure 5 wide-open canals. 

REPLY: enlarged canals usually referred to secondary remodeling. Here we just describe its shape. It should represent primary bone.

P16 paragraph 3 line 3. Compared to what?

REPLY: We have modified this sentence to make it more clear.” The vascular canals are longitudinally-oriented and relatively small in the outer cortex than in the mid and inner cortices”. 

P16 paragraph 3 line 5 Stay consistent in reporting percentages and report for all regions.

REPLY: We have added more percentages of cortical porosity in different parts of the cortex.

P16 paragraph 3 line 6 Also mention the LARGE one to the bottom of Fig. 6A.

REPLY: The large cavity to the bottom is a large erosional cavity surrounded by endosteal bone rather than secondary osteon because its size is much larger than other secondary osteons (See fig. S8). We have added the description of this large cavity. 

P17 line 1. 

Is this section truly from the mid-diaphysis? It looks to me like it's slightly towards the proximal or distal end which is fine, but do report it if that's the case. If it is in fact truly from the mid-diaphysis, the medullary cavity is much larger than in the other adult specimen and this difference should be discussed.

REPLY: Thank you for pointing out. The femur of IVPP V26546 that sampled only preserved the distal part. We rechecked this specimen and found that you are correct. The measurement of the preserved region is 5 cm, and the estimated whole length is about 14.6 cm. So this section was sampled near the distal region (See figure S1). We have added this information in the method and description part of the paper.

P19 Line 6.

Cite studies such as Köhler et al. 2012 that show LAGs are deposited even in fast growing animals in stressful enivornmental conditions.

REPLY: Cited as suggested.

P19 line13:

There are two discussion points missing from here I think. First, wouldn't the LAGs in the mid-cortex mean that Chinese Lystrosaurus was experiencing some kind of stressful season? Of anything, there is more indicators of stress in specimens from China than those from South Africa. I think if there is an argument to be made about the environment being more stressful in South Africa, you need more clarity on the size differences. If you can show that the South African specimens were smaller than your subadult specimens with LAGs then it would get around this issue. Second, I would move the section where species differences in bone tissue is discussed here and expand on differences that may be related to species-level differences. 

REPLY: we have added more comparison on body size between Africa and Chinese Lystrosaurus, and pointed out that our individuals are larger than sampled Africa Lystrosaurus from Early Triassic. We have also added species differences discussion here, although only a little could be clarified here. 

Page 19. Paragraph 2. Line5

Do you mean to say something more like, "an artifact of sampling only young individuals from South Africa"? This is not clearly written as is.

REPLY: Here we mean that Lystrosaurus from Early Triassic may die at a young age (indicated by fewer LAGs), so they are smaller than those from late Permian (should be at least subadult based on more LAGs). We have modified this sentence to “but this may the result of the young stage (indicates by rare LAGs) of the body in the Early Triassic based on bone microstructure”.

Page 19. Paragraph 2 line 6.

This formation has not been introduced. Describe it's age here for audience reference.

REPLY: Added as suggested. 

Page 20 line 10. 

Do you mean in the mid-cortex? This means something else in reference to the gross anatomy so I would avoid using it.

REPLY: We have modified this sentence to “---in the inner and mid cortex”

Page 20. Line 11. This jump is not logically laid out. You state a thick cortex and abundant canals suggests aquatic lifestyle. Then you state the cortical thickness (which I think needs clarification or remeasuring) but do not state whether that is thick or thin compared to other animals. You state that the ribs are different. And then you state that because of this, you can't say it's aquatic/semiaquatic. See general comments for further discussion.

REPLY: we have deleted this sentence and modified this paragraph properly to make it more clear. Most of our samples support Lystrosaurus has a thick cortex, as in Lystrosaurus of Africa. But thick cortex is not only seen in aquatic animals but also shown in extant burrowing animals. Therefore, thick cortex could not provide confident evidence of the aquatic or semiaquatic lifestyle of Lystrosaurus. The lifestyle of Lystrosaurus could not be certain right now, but we provided more methods to solve this problem. 

PAGE 28 

S1 Fig. I did not get this figure in my review document.

REPLY: Sorry. We have unloaded this figure in the revised version.

Figure 1. The guides in B to show were zoomed in photos are from are really helpful and necessary. I would suggest adding similar boxes to the rest of the figures for clarity.

REPLY: We have added boxes in other figures as suggested.

I would add anatomical directions (i.e. medial, lateral, posterior, anterior) in either the figure or the caption (e.g. medial to right, lateral to left, etc.). I would do this for all of the histology figures.

REPLY: Thank you for your suggestions. We have added anatomical directions in all the figures. 

Sincerely, 

Fenglu Han

---

## [Decision Letter · Decision Letter 1]

14 Dec 2020

PONE-D-20-17684R1

Bone histology of Lystrosaurus (Therapsida: Dicynodontia) from the Lower Triassic of North China, and its implication for lifestyle and environments after the end-Permian extinction

PLOS ONE

Dear Dr. Han,

Thank you for submitting your manuscript to PLOS ONE. After careful consideration, we feel that it has merit but does not fully meet PLOS ONE’s publication criteria as it currently stands. Therefore, we invite you to submit a revised version of the manuscript that addresses the points raised during the review process.

We look forward to receiving your revised manuscript.

Kind regards,

Jörg Fröbisch, Ph.D.

Academic Editor

PLOS ONE

Additional Editor Comments (if provided):

Dear authors,

given recent publications and difficulties in securing adequate reviewers during the first round of reviews, I requested additional reviews, the recommendations of which you can see below. I think that your study is well worthy of publication, but I strongly suggest that you carefully consider all comments and recommendations by the new reviewer, and modify your manuscript accordingly. I'm looking forward to receiving your revised manuscript again.

Best wishes,

Jörg Fröbisch

Reviewers' comments:

Reviewer's Responses to Questions

**Comments to the Author**

1. If the authors have adequately addressed your comments raised in a previous round of review and you feel that this manuscript is now acceptable for publication, you may indicate that here to bypass the “Comments to the Author” section, enter your conflict of interest statement in the “Confidential to Editor” section, and submit your "Accept" recommendation.

Reviewer #4: All comments have been addressed

Reviewer #5: (No Response)

2. Is the manuscript technically sound, and do the data support the conclusions?

Reviewer #4: Yes

Reviewer #5: No

3. Has the statistical analysis been performed appropriately and rigorously? 

Reviewer #4: N/A

Reviewer #5: N/A

4. Have the authors made all data underlying the findings in their manuscript fully available?

Reviewer #4: Yes

Reviewer #5: Yes

5. Is the manuscript presented in an intelligible fashion and written in standard English?

Reviewer #4: No

Reviewer #5: Yes

6. Review Comments to the Author

Reviewer #4: I appreciate all of the work put into this revision and find it greatly improved. I believe the paper now clearly demonstrates methodology and provides sufficient details to warrant publication. Particularly, S2 and S3 figs. Are well presented figures! I find it very helpful and thorough. There are only a few minor line edits that I found that only related to grammar and clarity. I have attached these comments in a marked version of the PDF.

An additional comment refers to S2 fig. The title of the figure says "accurate". I'm not sure what that is in reference to. But it would be better to have the limbs similarly oriented and the view explicitly stated (e.g. anterior).

Reviewer #5: This study describes the bone histology of seven Lystrosaurus individuals from the Lower Triassic of North China. The data have potential to provide interesting and new information on the growth patterns of Lystrosaurus as no studies have yet to focus on this taxon from China, potentially allowing for comparisons with Lystrosaurus from South Africa and Antarctica. However, given that the individuals could not be identified to species level (and thus there could be interspecific variation) I would call this study, at most, a preliminary work. They have very interesting conclusions suggesting that the Lystrosaurus from China are larger than those from the Triassic of South Africa and they lived longer (i.e. there are more LAGs in the Chinese specimens compared to the Triassic species from South Africa). This is the most interesting result of the study. However, I have some concerns relating to how much can be concluded from the data and these concerns should be addressed before the manuscript is ready for publication. My concerns are as follows.

1. Only three out of the seven individuals have more than one LAG – and the third element is a rib, which looks like it has a double LAG, not a triple LAG and certainly not an EFS. Double LAGs indicate one season (laid down in one year), so I would not call what you see in the rib as representing multiple seasons. That leaves the two femora – so only two individuals have three or four LAGs. I’m not convinced this is a large enough sample size to indicate that Chinese Triassic Lystrosaurus were growing for longer periods compared to South African Triassic species. At most this is a preliminary observation and more work needs to be done to see if you can replicate the data.

2. The data used to compare the size of Chinese Lystrosaurus with South African Lystrosaurus is outdated. Sam-pk-k8 and Sam-pk-8991 do not represent fully grown specimens or the largest known specimens. You can look at Botha 2020 (Botha, J. 2020. The paleobiology and paleoecology of South African Lystrosaurus. PeerJ 8:e10408. DOI: doi.org/10.7717/peerj.10408.). Granted, this publication came out while this study was under review and thus the authors have not had the opportunity to compare the data – but now is the time to do so. There is skull data for the maximum sizes of each South African species as well as size data for the limb bones used – i.e. the authors compare the size of the humerus, this can be done with both Triassic South African Lystrosaurus species to see how much larger the Chinese species is – if they really are notably larger and this can be confirmed, then it is a very interesting result, but this needs to be more adequately confirmed.

3. I disagree with the last ontogenetic stage being labelled adult. An adult will have a significant amount of slower growing tissues (parallel-fibered or lamellar) and/or an EFS – neither of which is present in the largest individuals. There is the beginning of parallel-fibered bone, so what the authors have captured here is the beginning of slower growth which would make these individuals subadults. It also cannot be confirmed if what is seen in the rib is an overall decreased growth or just parallel-fibered bone associated with a double LAG after which growth may have increased again. The two femora are undoubtedly older than the other individuals, but they are not adult. You could perhaps divide the stages into juvenile, early subadult and late subadult – although even saying late subadult is a push because there is so little parallel-fibered bone.

4. I have a problem with the methodology in calculating the cortical thickness of the bones. The problem with most dicynodonts is that they do not have a sharp transition from a clear open medullary cavity to a compact cortex. There is generally, and this is seen in Lystrosaurus, a gradual transition zone and there are often trabeculae within the medullary cavity. So if you are measuring the compact bone manually as the authors have done here, where do you objectively draw the line between compact bone and medullary cavity? It is very difficult to standardize and objectively measure cortical thickness in this way. This is why the program Bone Profiler is better than a manual procedure because the computer measures so many transects all around the bone. I don’t think any of the cortical thickness measurements are useful if they are done manually (unless you have sharp transition zones, which you don’t in Lystrosaurus). These calculations cannot be compared with other Lystrosaurus specimens or with other taxa. However, using Bone Profiler to calculate cortical thickness (which is generally K in the program) is a lot of work as the cross-sections needs to be changed into black and white images, which can only be done manually by colouring in the bone black and the spaces white – this is extremely time consuming. I thus, suggest that the entire cortical thickness section be deleted. The values cannot be reliably used and they don’t say much about the lifestyle of Lystrosaurus anyway. If the authors wish to keep the cortical thickness in the paper then they need to use Bone Profiler to obtain their values.

5. There is also a problem with the methodology for calculating cortical porosity. Fig S3 shows where the fields of view were taken – A, B and C are ok, but in D Field Of View 8, 9 and 10 are taken from the inner regions of the cortex, some of which includes resorption cavities – these cannot be compared with primary vasculature – so 8, 9 and 10 would give you inflated values compared to the others. Even in FOV 1-4 the FOV includes enlarged cavities and not just primary vasculature – this again inflates the values compared to 5-7. FOV is generally taken from the mid-cortex only and all 10 FOVs are taken from the “same” place giving you an average over 10 FOV. One can do what was done in A, B and C because each FOV is the same. E is also a problem because some FOVs are from the outer cortex, some from the middle and some from the inner regions – so you’re not standardizing where you’re taken the FOV from. Yes, you’re getting an average of the bone, but you’re doing it in a different way each time. F and particularly G I’m worried about as the inner regions of the FOV are resorption cavities which you cannot include in primary vasculature – so here, the entire bone’s values are inflated, same with H – in fact in H, there is even medullary cavity, so was this included as space or was it ignored? If the latter then your FOV is smaller depending on where the FOV was taken, so you’re not comparing same-sized FOVs. This is going to produce unreliable results.

6. It would also be good to include Whitney, M.R. and Sidor, C.A. 2020. Evidence of torpor in the tusks of Lystrosaurus from the Early Triassic of Antarctica. Communications Biology 3:471. DOI: 10.1038/s42003-020-01207-6. Although they do not deal with limb bones (they use teeth) they do talk about different environments affecting Lystrosaurus. This publication may have come out while this study was still in review – but it would be good to include it now for a discussion on differing environments affecting Lystrosaurus growth.

I have made numerous comments on the actual manuscript (on the tracked changes section), too many to mention here, but these queries should also be addressed before the manuscript can be published.

7. PLOS authors have the option to publish the peer review history of their article (what does this mean?). If published, this will include your full peer review and any attached files.

Reviewer #4: **Yes: **Megan Whitney

Reviewer #5: **Yes: **Jennifer Botha

---

## [Author Response · Author response to Decision Letter 1]

6 Jan 2021

Jan. 6th, 2021

Dear Editor and Reviewers:

Thank you for your letter and for the reviewers’ comments concerning our manuscript entitled “Bone histology of Lystrosaurus (Therapsida: Dicynodontia) from the Lower Triassic of North China, and its implication for lifestyle and environments after the end-Permian extinction” (ID: PONE-D-20-17684). Those comments are all valuable and very helpful for revising and improving our paper, as well as important for highlighting the significance of our work. We have studied the comments carefully and made a number of corrections to address them, which we hope to meet with approval. The revised portions of the MS were marked by red colors. The main corrections to the paper and the responses to the editor and reviewer’s comments are as following:

Reviewer #4: I appreciate all of the work put into this revision and find it greatly improved. I believe the paper now clearly demonstrates methodology and provides sufficient details to warrant publication. Particularly, S2 and S3 figs. Are well presented figures! I find it very helpful and thorough. There are only a few minor line edits that I found that only related to grammar and clarity. I have attached these comments in a marked version of the PDF.

REPLY: Thank you for all your comments and corrections. All the minor comments in the pdf file have been corrected. In addition, the revised version has also been modified a lot based on another reviewer’s comments. For example, we have deleted some values of cortical thickness and cortical porosity that were thought to be objective, but we still keep S2 and S3 Figs that can show bone microstructure more clear. We also have added new comparisons between Chinese and South Africa Lystrosaurus based on the new published paper by Botha et al. (2020). We hope all these modifications are comfortable for you. 

An additional comment refers to S2 fig. The title of the figure says "accurate". I'm not sure what that is in reference to. But it would be better to have the limbs similarly oriented and the view explicitly stated (e.g. anterior).

REPLY: I think you mean S1 Fig. We just pointed the positions that we sampled in the figure. We have removed the word “accurate” and added limb orientation as suggested.

Reviewer #5: This study describes the bone histology of seven Lystrosaurus individuals from the Lower Triassic of North China. The data have potential to provide interesting and new information on the growth patterns of Lystrosaurus as no studies have yet to focus on this taxon from China, potentially allowing for comparisons with Lystrosaurus from South Africa and Antarctica. However, given that the individuals could not be identified to species level (and thus there could be interspecific variation) I would call this study, at most, a preliminary work. They have very interesting conclusions suggesting that the Lystrosaurus from China are larger than those from the Triassic of South Africa and they lived longer (i.e. there are more LAGs in the Chinese specimens compared to the Triassic species from South Africa). This is the most interesting result of the study. However, I have some concerns relating to how much can be concluded from the data and these concerns should be addressed before the manuscript is ready for publication. My concerns are as follows. 

1. Only three out of the seven individuals have more than one LAG – and the third element is a rib, which looks like it has a double LAG, not a triple LAG and certainly not an EFS. Double LAGs indicate one season (laid down in one year), so I would not call what you see in the rib as representing multiple seasons. That leaves the two femora – so only two individuals have three or four LAGs. I’m not convinced this is a large enough sample size to indicate that Chinese Triassic Lystrosaurus were growing for longer periods compared to South African Triassic species. At most this is a preliminary observation and more work needs to be done to see if you can replicate the data.

REPLY: We agree that our work is just a preliminary observation and much work needs to be done in the future, but we do not think the third element only has a double LAG that represents one year. The third element is a rib, which has three LAGs that were identified previously. We agree that only two prominent LAGs are shown in Fig. 6J, and the third outermost LAG is weak. At least, there are two LAGs present. We supposed it to be two LAGs rather than a double LAG, mainly based on the following evidence: these two LAGs are isolated, vary out of sync, and never combined; the related femur has more than 2 LAGs and a wide parallel-fibered bone at the periphery, which was sampled by another researcher and will be published elsewhere so that could not use in this paper, unfortunately. Furthermore, the LAGs are more closed to each other in the thin cortex (such as ribs, fibula, etc) than in the robust cortex (such as humerus, femur and tibia). For example, see the paper Han et al (2020, the bone histology of Jeholosaurus). The distinct between LAGs is much narrower in the fibula than that in the tibia. Finally, this specimen (sampled rib) is the largest individual that we sampled and it is reasonable that it has more LAGs than others. 

2. The data used to compare the size of Chinese Lystrosaurus with South African Lystrosaurus is outdated. Sam-pk-k8 and Sam-pk-8991 do not represent fully grown specimens or the largest known specimens. You can look at Botha 2020 (Botha, J. 2020. The paleobiology and paleoecology of South African Lystrosaurus. PeerJ 8:e10408. DOI: doi.org/10.7717/peerj.10408.). Granted, this publication came out while this study was under review and thus the authors have not had the opportunity to compare the data – but now is the time to do so. There is skull data for the maximum sizes of each South African species as well as size data for the limb bones used – i.e. the authors compare the size of the humerus, this can be done with both Triassic South African Lystrosaurus species to see how much larger the Chinese species is – if they really are notably larger and this can be confirmed, then it is a very interesting result, but this needs to be more adequately confirmed.

REPLY: Thank you for your new information. We have added more comparison of the known humeral length with a LAG between Chinese Lystrosaurus and South Africa from Early Triassic, we found that their size has no significant difference (S4 Fig). But Chinese Lystrosaurus still has a relatively large proportion with LAGs compared to South Africa Lystrosaurus. The humeral length larger than 10 cm is the most common size based on our observation. Both the body size and bone microstructure support that many Lystrosaurus have reached at least the subadult stage, unlike Triassic Lystrosaurus from Africa where subadults are rare. Of Course, we need more work to clarify this conclusion. 

3. I disagree with the last ontogenetic stage being labelled adult. An adult will have a significant amount of slower growing tissues (parallel-fibered or lamellar) and/or an EFS – neither of which is present in the largest individuals. There is the beginning of parallel-fibered bone, so what the authors have captured here is the beginning of slower growth which would make these individuals subadults. It also cannot be confirmed if what is seen in the rib is an overall decreased growth or just parallel-fibered bone associated with a double LAG after which growth may have increased again. The two femora are undoubtedly older than the other individuals, but they are not adult. You could perhaps divide the stages into juvenile, early subadult and late subadult – although even saying late subadult is a push because there is so little parallel-fibered bone.

REPLY: We have changed the word “adult” to “late subadult” stage as suggested.

4. I have a problem with the methodology in calculating the cortical thickness of the bones. The problem with most dicynodonts is that they do not have a sharp transition from a clear open medullary cavity to a compact cortex. There is generally, and this is seen in Lystrosaurus, a gradual transition zone and there are often trabeculae within the medullary cavity. So if you are measuring the compact bone manually as the authors have done here, where do you objectively draw the line between compact bone and medullary cavity? It is very difficult to standardize and objectively measure cortical thickness in this way. This is why the program Bone Profiler is better than a manual procedure because the computer measures so many transects all around the bone. I don’t think any of the cortical thickness measurements are useful if they are done manually (unless you have sharp transition zones, which you don’t in Lystrosaurus). These calculations cannot be compared with other Lystrosaurus specimens or with other taxa. However, using Bone Profiler to calculate cortical thickness (which is generally K in the program) is a lot of work as the cross-sections needs to be changed into black and white images, which can only be done manually by colouring in the bone black and the spaces white – this is extremely time consuming. I thus, suggest that the entire cortical thickness section be deleted. The values cannot be reliably used and they don’t say much about the lifestyle of Lystrosaurus anyway. If the authors wish to keep the cortical thickness in the paper then they need to use Bone Profiler to obtain their values.

REPLY: Thank you for your suggestions. In fact, we tried to use the software bone profiler after your suggestion in your first review. But we found many problems and we gave up finally. First, we did not find a valid link that could download this software. The official link provides a new version that analyses on the website but was also not available for analysis right now. In addition, I do not quite understand the principle of the software. All the bone has the same color, then how to separate compact bone and cancellous bone in the program? Girondot and Laurin (2003, p460, secondary paragraph) mentioned that K was the ratio between the internal and external diameter of the bone, so it should not represent the cortical thickness that we mentioned here.

Finally, we agree that a gradual transition from compact bone to medullary cavity has happened in Lystrosaurus, but not all specimens are difficult to separate. There is still a clear change in some specimens, such as the femur IVPP V26542. This situation is also mentioned by Ray et al (2005). So in the revised version, we keep the cortical thickness values that we can discern the transition clearly and have omitted other values with a vague transition between compact bone and medullary region. Furthermore, we have also omitted cortical porosity values with vague transitions. 

5. There is also a problem with the methodology for calculating cortical porosity. Fig S3 shows where the fields of view were taken – A, B and C are ok, but in D Field Of View 8, 9 and 10 are taken from the inner regions of the cortex, some of which includes resorption cavities – these cannot be compared with primary vasculature – so 8, 9 and 10 would give you inflated values compared to the others. Even in FOV 1-4 the FOV includes enlarged cavities and not just primary vasculature – this again inflates the values compared to 5-7. FOV is generally taken from the mid-cortex only and all 10 FOVs are taken from the “same” place giving you an average over 10 FOV. One can do what was done in A, B and C because each FOV is the same. E is also a problem because some FOVs are from the outer cortex, some from the middle and some from the inner regions – so you’re not standardizing where you’re taken the FOV from. Yes, you’re getting an average of the bone, but you’re doing it in a different way each time. F and particularly G I’m worried about as the inner regions of the FOV are resorption cavities which you cannot include in primary vasculature – so here, the entire bone’s values are inflated, same with H – in fact in H, there is even medullary cavity, so was this included as space or was it ignored? If the latter then your FOV is smaller depending on where the FOV was taken, so you’re not comparing same-sized FOVs. This is going to produce unreliable results.

REPLY: We agree that there are still some problems with measuring cortical porosity. The best way is to measure the whole compact bone. However, there are at least two problems: As we mentioned above, there is no sharp transition between compact bone and medullary cavity in some specimens, and the areas that we sampled may be objective. Therefore, in the revised version, we only keep the cortical porosity values with clear tissue change. Secondly, some bone sections do not show clear bone tissue in some parts due to bad preservation, such as Fig S3 E. It has a clear medullary cavity, but not allowed us to standard the acute position. It just provides a reference, that is why we mentioned that it was an estimated value in the method. Nevertheless, we have omitted this value according to the reviewer’s comments.

6. It would also be good to include Whitney, M.R. and Sidor, C.A. 2020. Evidence of torpor in the tusks of Lystrosaurus from the Early Triassic of Antarctica. Communications Biology 3:471. DOI: 10.1038/s42003-020-01207-6. Although they do not deal with limb bones (they use teeth) they do talk about different environments affecting Lystrosaurus. This publication may have come out while this study was still in review – but it would be good to include it now for a discussion on differing environments affecting Lystrosaurus growth.

REPLY: We have added this reference and mentioned that environments can affect tusk and bone microstructure of Lystrosaurus. But we did not discuss a lot because it was difficult for comparison between teeth and bone microstructure.

Comments in the pdf

P5, line 10

Comment: you need to explain the difference between a LAG and an annulus. i.e. a LAG is a cement line and indicates a temporary, but complete cessation in growth whereas an annulus consists of lamellar or parallel-fibered bone and indicates a temporary decrease in growth rate

REPLY: We have added this information.

Table 1. Comment: fix this column so that the s is not on another line

REPLY: Modified as suggested.

P6, line 2

why was this specimen chosen? is it the largest known in China? is it the most complete? does it have a skull? can you identify it to species level? You say that the ontogenetic stage is unknown - but is it small or big? are sutures fused? are their muscle scars? etc you must be able to estimate the ontogenetic stage of this specimen. Are you estimating all these other values based on a juvnenile or an adult?

REPLY: we have added more information on this specimen in the method. It is the holotype of L. hedini and represents the most complete specimen of Lystrosaurus (Young 1935). It has a skull length of 173 mm and includes most of the postcranial material. It is probably a subadult individual based on open neurocentral sutures and fused sacral vertebrae. It has a humeral length of 11.5 cm, which is more suitable to assign it to be within the subadult stage compared to others.

P6, paragraph 1

Comment: This is an outdated method for calculating cortical thickness and is problematic in an animal such as Lystrosaurus because there is no clear strict transition between cortical bone and medullary cavity. The broad transition zone in this region makes measurements calculating cortical thickness quite subjective. I recommend using Bone Profiler for this method. These measurements will then be more useful in that they can be compared with other studies.

REPLY: We have already given a detailed explanation above. We have omitted controversial data but keep some measurements that have a clear transition. As we provided the accurate position that we measured (Fig. S2). The reader can choose whether to use it or not. 

P6, paragraph 1, Line 11

This method differs from that of previous studies - which take 10 fields of view from the mid-cortex all around the cross-section. It is unclear if you are doing a transect from the inner to outer cortex and getting the average, or if you are taking inner versus outer fields of view randomly all over the cross-section - if the latter then this method is not being very consistent. You need to standardize this method more.

REPLY: We have given a detailed response above. In addition, we are doing an average cortical porosity from the inner to outer cortex rather than the mid cortex. As we know that cortical porosity is different from the inner to outer cortex, so it also has a problem to just choose mid-cortex, because cortical porosity is gradually changed from the inner to outer cortex. How do we define the mid cortex is a problem and objective. 

P7, paragraph 2. 

Comment: There is a new reference which will help you compare your results to the SA Lystrosaurus, you need to include this:

Botha, J. 2020. The paleobiology and paleoecology of South African Lystrosaurus. PeerJ 8:e10408. DOI: doi.org/10.7717/peerj.10408.

REPLY: Thank you for your information. We have added this reference and more comparison in the discussion part.

P7, table 2

Comment: separate into two separate columns

REPLY: modified as suggested.

P7, table 2

Comment: so would these values mark the highest being the inner cortex and the lowest value being the outer cortex?

REPLY: Sorry for the misunderstanding. Providing the variation of cortical thickness was suggested by Megan Whitney. It is the ratio of compact bone (one side) to the diameter of the whole cross-section. For example, see Fig S2E. The dorsal cortex is thin and the ventral cortex is relatively thick, so the cortical thickness ranges from 16-41%. Please see the method part for detailed information. 

P8, table 2

extensive secondary reconstruction would imply dense Haversian bone, or that the reconstruction has reached the outer cortex - is this the case for your elements here? It doesn't look like it from the figures

REPLY: Thank you for pointing it out. We have modified this sentence as “a few secondary osteons”

P8, Juvenile, line 5.

Comment: where would you denote the edge of the medullary cavity here with such a gradual transition zone? You need to explain how you got these measurements and how you ensured that they were not subjective, but standardized

REPLY: We have added more explanation on how to identify compact bone and medullary cavity. The compact bone tissues are connected tightly without bone trabecula and large resorbed cavity, whereas the medullary cavity either empty (we call it free medullary cavity) or filled with trabecula, which is usually isolated or loosely connected. However, sometimes, the trabecula is abundant and show a similar connection as in compact bone, and could not be separated clearly. 

P9 line 2. Comment: in circumferential rows in parts?

REPLY: Yes, added.

P9, line 11. 

not shown in Fig 1B - label or take out in the text here

REPLY: The word “Fig. 1B” has been taken out here.

P9, figure 1. I know what you mean here by wide-open canals - but this isn't a very good term. Just label them large primary canals because that's what they are.

REPLY: Thank you for your suggestions. Modified.

P9, last line.

Comment: then how did you calculate cortical thickness?

P10 line 8. Comment: then how accurate is the cortical thickness measurement?

REPLY: We have provided more explanation on defining medullary cavity and compact bone in the method, See our explanation above. 

P10, Fig 2 caption N.

Comment: I'm not convinced those are secondary osteons - I cannot see a cement line or How ship's lacunae - you need a better close up of this

REPLY: We have changed to a close up new picture to show secondary osteon here. 

P11 paragraph 2 line 11.

Comment: according to your definition then - these growth marks cannot be seen around the whole circumference of the bone? So are they really indicative of a change in growth rate?

REPLY: These growth marks usually associated with parallel-fibered bone in the outermost region, so we consider it to indicate a slow growth rate. 

P12 Subadult. Line 11.

Comment: This is not shown from Fig 3A and B - I would not call those channels "significantly reduced"

REPLY: we have removed the word “significantly”. 

P12 Subadult. Line 12. can this LAG be followed around the whole circumference?

REPLY: This LAG is weak and not clear in some parts under normal light, and it is also not clear under polarized light, so we revised it to be a growth mark in the revised MS. Therefore, we remove this individual to be within a juvenile stage.

P13. Line 1. cannot see the LAG here

REPLY: We have removed the word “a LAG” here.

P13 paragraph 2 Line 13 

It looks like a double LAG - don't say several otherwise it sounds like an EFS.

REPLY: We did not say several LAGs. There is only one prominent LAG and we wrote “a distinct LAG comprised of several growth marks”. It does show more than two growth marks (weak growth lines) but not LAGs. We remarked the arrow positions to show growth marks that more clear.

P13 Figure 4 caption. Line 3. 

Comment: I'm not convinced - looks like two not three. And does the multiple number stay consistent around the whole cortex or does it become one LAG at some point?

REPLY: Again, we did not say it has three LAGs, instead, it is just one LAG. But we can see that it contains several growth marks. It becomes narrower but still a distinct LAG in other regions. 

P13 Figure 4 caption. Line 5. 

Comment: this looks like a single LAG here

REPLY: Yes. We agree. It is a single LAG containing several growth marks (or weak growth lines). 

P14 line 1. Comment: looks like a double LAG here too

REPLY: Yes. We agree and have modified in the revised MS. 

P14 line 1. this isn't a normal term - they are either resorption canals or canals that are just starting to form - look to see which one

REPLY: Yes. Here we only describe its shape that was suggested by another reviewer. They probably represent resorption canals.

P14 paragraph 2 line 6

Comment: so what are these? how far do they go around the cortex? do they run with the LAG or are they separate?

REPLY: Here we mean that a clear LAG was formed by several growth marks (weak growth lines that vary throughout the whole cortex), but we rethink parallel-fibered bone is more suitable. We have modified this sentence as “A clear LAG, as well as parallel-fibered bone, are present at the mid-region as in the fibula”.

P14 paragraph 2 line 7

Comment: cannot still be abundant but also strongly reduced - rephrase - I wouldn't call the canals strongly reduced. Perhaps there is a slight decrease in vascular canal size towards the periphery

REPLY: We have changed the word “strongly” to “gradually”.

P14 adult stage. 

Comment: how do you define adult tissues?

REPLY: Adult stage was usually defined by the presence of EFS or lamellar/parallel-fibered bone tissue at the periphery. Here, we found parallel-fibered bone at the periphery. However, we agree that the layer of parallel fibered bone was still narrow. We have changed this stage to “late subadult” as suggested.

P15 Line 1 

Comment: as with my comments above for every element - how then do you decide where the edge of the compact cortex is?

REPLY: cortical thickness and porosity have been removed in this specimen.

P15 paragraph 1 last line. 

Comment: you've added in Placerias - but an EFS has been found in Kannemeyeria as well - see Botha-Brink and Angielczyk 2010 - those are the only two so you may as well name them seen as they are dicynodonts

REPLY: We have added this reference. 

P15 Fig 5 caption. 

Comment: I am not convinced this is an adult - at most, with such a small region of parallel-fibered bone at the periphery we are seeing a first transition to slower forming bone tissues - so at most this is a subadult. An adult would have a much larger region or parallel-fibered bone or lamellar bone or an EFS

REPLY: We have changed the stage “adult” to “late subadult”.

P15 Last line. 

Comment: change this term

REPLY: Modified. 

P16 Fig. 6. Comment: looks like a double LAG not triple

REPLY: This should be at least two LAGs rather than a double LAG. These two LAGs have more than three LAGs and a wide parallel fibered bone have been founded in the associated femur. But it has already been sampled by another research and will be published elsewhere. 

P17. Line 5 I would say two

REPLY: Modified as suggested.

P17. Line 5. 

Comment: no - not enough to indicate an EFS - looks like a double LAG. This is not an EFS - as you decide it's not one from your next comment - rather just take out the phrase "similar to an EFS"

REPLY: We have deleted this sentence. 

P17 Discussion. Line 4. 

Comment: There is not enough evidence to indicate this third stage - the rib cannot be used for this and the two femora look like subadults

REPLY: We have modified the “adult” to “late subadult” and modified the text to make it more clear.

P17 Discussion. Line 6. 

no - you don't know that - no EFS - a double LAG is not enough to indicate general slow growth - definite subadult, but I wouldn't call these femora and rib adults

REPLY: We have changed the “adult” to “late subadult” stage. 

P17 Discussion. Line 6.

Comment: what about the humerus?

REPLY: We have added the word“the humerus”.

P18 Paragraph 3 line 4. 

Comment: only three out of seven have more than 1 LAG - and the rib looks like a double LAG - i.e. indicating one season, the rib is not a good one to include in stating that more than 1 or 2 LAGs are present. That just leaves the two femora that have 3 or 4 LAGs - this is not a high sample size

REPLY: As we mentioned above, the rib in Figure 6 has at least two LAGs, and this individual also sampled the femur which has more than 2 LAGs and wide parallel-fibered bone tissue at the periphery. But it has been sampled by another researcher and will be published elsewhere.

P18 Paragraph 3 line 8

these are not the largest Triassic Lystrosaurus - compare your elements with Botha 2020

P18 Paragraph 3 line 9

confirm this by comparing with the largest Triassic humeri in Botha 2020

REPLY: We have added a new comparison and S4 Figure according to this new reference, and we found that there is no significant size difference between Chinese and South Africa Lystrosaurus, but Chinese Lystrosaurus tends to have more LAGs, which may be affected by different species or environment. This did not affect our conclusion that Chinese Lystrosaurus have reached at least the subadult stage based on the presence of LAGs and relatively large body size (see revised MS for detailed information). 

P19. Line 3.

Comment: you don't show this in the table - in the table it's just the two femora that have more than two LAGs. There is only one juvenile humerus in your tables

REPLY: We have modified this sentence as “All the other individuals with humeral length larger than 12 cm (including estimated value) have more than two LAGs to be among late subadult stage (Table 1)”. The two femora IVPP V26546 and IVPP V26547 have more than two LAGs. IVPP V26546 also preserves a humerus (12 cm), although we did not sample it. IVPP V26547 and IVPP V26548 do not preserve the humeri, but the estimated humeral length is larger than 12 cm. We have changed the “Elements” to “Sampled elements” to make it more clear. 

P20 Line 1

Comment: yes but this has been shown to be false in Botha 2020 - the size reduction is just because they didn't live long enough to attain large sizes, not because of miniaturization.

REPLY: We have added this new reference. 

P20 paragraph 2 line 7

Comment: cannot start a sentence with and

REPLY: Modified. 

P20 paragraph 2, line 8

Comment: not necessarily - it depends on where you sampled your rib. Cortical thickness in ribs differ along the length of the shaft and you haven't said where your sample was taken from - so it's very difficult to compare ribs unless you know for sure they were sampled at the same position along the shaft

REPLY: We have deleted this sentence.

Sincerely, 

Fenglu Han

---

## [Decision Letter · Decision Letter 2]

1 Feb 2021

PONE-D-20-17684R2

Bone histology of Lystrosaurus (Therapsida: Dicynodontia) from the Lower Triassic of North China, and its implication for lifestyle and environments after the end-Permian extinction

PLOS ONE

Dear Dr. Han,

Thank you for submitting your manuscript to PLOS ONE. After careful consideration, we feel that it has merit but does not fully meet PLOS ONE’s publication criteria as it currently stands. Therefore, we invite you to submit a revised version of the manuscript that addresses the points raised during the review process.

We look forward to receiving your revised manuscript.

Kind regards,

Jörg Fröbisch, Ph.D.

Academic Editor

PLOS ONE

Additional Editor Comments (if provided):

Dear authors,

please find attached another round of comments and recommendations by one of the reviewers, Jennifer Botha. Please carefully address ALL points made by her. Based on the very limited dataset, I further strongly recommend that you indicate the preliminary nature of this study in the title of the manuscript by changing the title to something like: "Preliminary bone histological analysis of Lystrosaurus (Therapsida: Dicynodontia) from the Lower Triassic of North China, and its implication for lifestyle and environments after the end-Permian extinction".

I'm looking forward to your revised version!

Thanks and best wishes,

Jörg Fröbisch

Reviewers' comments:

Reviewer's Responses to Questions

**Comments to the Author**

1. If the authors have adequately addressed your comments raised in a previous round of review and you feel that this manuscript is now acceptable for publication, you may indicate that here to bypass the “Comments to the Author” section, enter your conflict of interest statement in the “Confidential to Editor” section, and submit your "Accept" recommendation.

Reviewer #5: (No Response)

2. Is the manuscript technically sound, and do the data support the conclusions?

Reviewer #5: Partly

3. Has the statistical analysis been performed appropriately and rigorously? 

Reviewer #5: N/A

4. Have the authors made all data underlying the findings in their manuscript fully available?

Reviewer #5: Yes

5. Is the manuscript presented in an intelligible fashion and written in standard English?

Reviewer #5: Yes

6. Review Comments to the Author

Reviewer #5: I have reviewed a revision of this manuscript and it is greatly improved. The authors have clarified or corrected most issues, but I still have a few queries that perhaps need further clarification.

1. The main issue I have pertains to how the authors define growth marks. Growth marks, growth lines and growth rings all mean the same thing, and include both LAGs and annuli. A LAG is a cement line, and there can be more than one closely spaced LAG, 2 LAGs is referred to as a double LAG, 3 as triple etc. When the LAGs are so closely spaced that there are no vascular canals between them, osteohistologists refer to these growth marks as double or triple LAGs etc – they pertain to one season (or one year), and so are not counted separately (as in a triple LAG would mean a one year old not a three year old). The authors sometimes refer to a LAG containing several growth marks. This is confusing because a LAG is a growth mark and a double or triple LAG is referred to as such, not a single LAG consisting of several growth marks. This needs a bit of clarification in the text (which I note in my comments on the pdf). (e.g. early subadult 26545 fibula is described as have a LAG of several growth marks),

2. The LAG number in juvenile humerus IVPP V26544 is unclear. In the table the authors say 0 growth rings, but in the text they note a LAG. I don’t know if this is because of issue 1 where there needs to be clarification about the definition, or whether the table needs to be changed to show that there is a growth mark in this bone.

3. In the discussion the authors mention the larger body size of the Chinese Lystrosaurus compared to the South African species but the graph in S4 shows the Chinese Lystrosaurus to be equal to the Triassic species of South Africa, I don’t see how the Chinese Lystrosaurus are larger.

4. The authors suggest that there were better conditions in North China compared to South Africa, but the Chinese specimens have more LAGs and they don’t seem to be any bigger than the SA specimens. If the Chinese specimens have more LAGs wouldn’t this rather suggest that the China environment was more harsh (causing them to react to a more seasonal environment and deposit LAGs), and/or that the Chinese specimens were stunted if they have more LAGs (and thus older) but are not physically larger?

I have made a few comments on the actual manuscript (on the tracked changes section). Once these issues have been clarified, I think the manuscript is ready for publication.

7. PLOS authors have the option to publish the peer review history of their article (what does this mean?). If published, this will include your full peer review and any attached files.

Reviewer #5: **Yes: **Jennifer Botha

---

## [Author Response · Author response to Decision Letter 2]

10 Feb 2021

Dear Editor and Reviewers:

Thank you for your letter and for the reviewers’ comments concerning our manuscript entitled “Bone histology of Lystrosaurus (Therapsida: Dicynodontia) from the Lower Triassic of North China, and its implication for lifestyle and environments after the end-Permian extinction” (ID: PONE-D-20-17684). Those comments are all valuable and very helpful for revising and improving our paper, as well as important for highlighting the significance of our work. We have studied the comments carefully and made a number of corrections to address them, which we hope to meet with approval. The revised portions of the MS were marked by red colors. The main corrections to the paper and the responses to the editor and reviewer’s comments are as following:

Editor’s comment: I further strongly recommend that you indicate the preliminary nature of this study in the title of the manuscript by changing the title to something like: "Preliminary bone histological analysis of Lystrosaurus (Therapsida: Dicynodontia) from the Lower Triassic of North China, and its implication for lifestyle and environments after the end-Permian extinction".

REPLY: We have modified the title as suggested. 

Reviewer #5: I have reviewed a revision of this manuscript and it is greatly improved. The authors have clarified or corrected most issues, but I still have a few queries that perhaps need further clarification.

1. The main issue I have pertains to how the authors define growth marks. Growth marks, growth lines and growth rings all mean the same thing, and include both LAGs and annuli. A LAG is a cement line, and there can be more than one closely spaced LAG, 2 LAGs is referred to as a double LAG, 3 as triple etc. When the LAGs are so closely spaced that there are no vascular canals between them, osteohistologists refer to these growth marks as double or triple LAGs etc – they pertain to one season (or one year), and so are not counted separately (as in a triple LAG would mean a one year old not a three year old). The authors sometimes refer to a LAG containing several growth marks. This is confusing because a LAG is a growth mark and a double or triple LAG is referred to as such, not a single LAG consisting of several growth marks. This needs a bit of clarification in the text (which I note in my comments on the pdf). (e.g. early subadult 26545 fibula is described as have a LAG of several growth marks),

REPLY: Thank you for clarifying the meaning of these words. In our paper, LAGs, growth lines (or rings) are the same things that indicate seasonal change, but growth marks are a little different. We follow Horner etal (1999) and Green etal (2010) in only identifying growth marks as LAGs and annuli if they are traced around the full circumference of the bone. Therefore, one growth mark may not represent one year, unlike one LAG. I agree entirely with your definition of double LAG and triple LAGs. In the new revised version, we have added the definition of growth marks in the method. Moreover, we have removed the statement of “ a LAG containing several growth marks”.

2. The LAG number in juvenile humerus IVPP V26544 is unclear. In the table the authors say 0 growth rings, but in the text they note a LAG. I don’t know if this is because of issue 1 where there needs to be clarification about the definition, or whether the table needs to be changed to show that there is a growth mark in this bone.

REPLY: As we have mentioned above, the growth mark in the text is not a true LAG, for it is unstable and does not go across the whole cross-section of the bone. Here we assume that a growth mark does not represent one year.

3. In the discussion the authors mention the larger body size of the Chinese Lystrosaurus compared to the South African species but the graph in S4 shows the Chinese Lystrosaurus to be equal to the Triassic species of South Africa, I don’t see how the Chinese Lystrosaurus are larger.

REPLY: Sorry for the confusion. In S4, the graph only compares Lystrosaurus with LAGs. But most of South Africa Lystrosaurus do not have LAGs and have small sizes (this can be checked from the measurement of Lystrosaurus in Botha et al. 2020). Therefore, generally, the Chinese Lystrosaurus looks larger because most of them reached subadult here. As this is a little confusing, we have deleted “large body size” in the revised version. 

4. The authors suggest that there were better conditions in North China compared to South Africa, but the Chinese specimens have more LAGs and they don’t seem to be any bigger than the SA specimens. If the Chinese specimens have more LAGs wouldn’t this rather suggest that the China environment was more harsh (causing them to react to a more seasonal environment and deposit LAGs), and/or that the Chinese specimens were stunted if they have more LAGs (and thus older) but are not physically larger?

REPLY: This is a very good question. Our conclusion that North China may have a better environmental condition than South Africa is mainly based on most Chinese Lystrosaurus has reached subadult stage when they died, whereas most South Africa Lystrosaurus are still young when they died. The “high juvenile excess mortality” (Botha et al. 2020) should be caused by an unusual environmental condition, although we still do not know the accurate condition. The presence of LAGs with large size suggests that the Chinese Lystrosaurus is at least reach a subadult stage. They have a relatively better condition than South Africa, but it does not mean that they have a good environmental condition. The environment may be harsh, but they could survive to a late stage may be through seasonal reduction in metabolic activity (torpor), like Lystrosaurus from Antarctica (Whitney and Sidor 2020). We have added this new statement. 

Whether the Chinese Lystrosaurus have more LAGs than those from South Africa with similar sizes still needs more data to prove. In addition, the presence of LAGs was also affected by different species. For example, I notice that L. declivis with one LAG is large than L. murrayi (figure S4). But this needs more data to support. 

Comments in the PDF

Page 3, line 9. you should add in here, but see Botha 2020

REPLY: Added as suggested.

Page 9. Table 1. Comment: take this out - I don't see an e anywhere

REPLY: Deleted.

Page 11 line 6. but in the table you say 0 growth rings

Page 12 line 1. Comment: as growth mark number is important in this study you need to say how many growth marks are present. You only show one in 2G and H, but you mention growth marks plural - and in the table you say 0 growth rings, I think there is an issue here on your definition of growth marks

REPLY: We have modified “growth marks” to “one growth mark”. As we mentioned above, the growth mark could not trace around the whole cross-section of bone and may not indicate a year (unlike LAG or growth ring), so we wrote 0 growth ring in the table. 

Page 13. Fig. 3 line 3. I would call this a LAG

REPLY: Here the picture shows the most prominent line, but it is only clear in this region and not shown in other parts of the bone, so we just say it is a growth mark, which has already been defined in the method part. 

Page 14. Line 2. This is where the definition of growth marks is an issue - you show 3 arrows very close together, with no vascular canals between them - each growth mark here is a LAG - so this forms a triple LAG - which is interesting because triple LAGs have not been found in the SA Triassic species - but it needs to be remembered that this still represents only one season

REPLY: Here we prefer not to use the word “a triple LAG” because only one prominent LAG go across the whole cross-section, other lines disappeared soon. In addition, the third rest line is too weak and we have modified as “the presence of two rest lines” here. Therefore, the revised sentence is “A distinct LAG is present near the mid-region of the cortex (Fig 4A), and it separates into two closely spaced rest lines in some regions (Fig 4B).”

Page 14. Fig. 4. Line 3. you need to rephrase - this is a fibula showing a triple LAG

REPLY: We have modified this sentence as “partial outer cortex of the fibula showing two closely spaced rest lines (arrows).”

Page 14. Fig. 4. Line 7. Rephrase to a double LAG

REPLY: Modified as suggested. This is a double LAG as they are very stable and go across the whole cross-section.

Page 16 paragraph 2, line 7. 

Comment: if it's "a" avascular layer - then the LAG is accompanied by an annulus - not annuli

REPLY: We have modified this sentence as “A LAG with a narrow avascular layer of an annulus is present in the mid-cortex. 

Page 16 paragraph 2, line 9.

this like C shows one annulus - not annuli

REPLY: Corrected.

Page 19, paragraph 2, line 1. 

Comment: but your graph shows that the Chinese Lystrosaurus is similar in size to the of the Triassic South African sample

REPLY: Right. But the graph only compares Lystrosaurus with LAGs. But most of South Africa Lystrosaurus do not have LAGs and have small sizes. Therefore, generally, the Chinese Lystrosaurus is larger. Here we have deleted “large body size” for this confusion. 

All other grammar mistakes that mentioned have been corrected. 

Thank you and best regards, 

Fenglu Han

---

## [Editor Report · Decision Letter 3]

4 Mar 2021

Preliminary bone histological analysis of Lystrosaurus (Therapsida: Dicynodontia) from the Lower Triassic of North China, and its implication for lifestyle and environments after the end-Permian extinction

PONE-D-20-17684R3

Dear Dr. Han,

We’re pleased to inform you that your manuscript has been judged scientifically suitable for publication and will be formally accepted for publication once it meets all outstanding technical requirements.

Kind regards,

Jörg Fröbisch, Ph.D.

Academic Editor

PLOS ONE
---

## [Editor Report · Acceptance letter]

11 Mar 2021

PONE-D-20-17684R3 

Preliminary bone histological analysis of *Lystrosaurus* (Therapsida: Dicynodontia) from the Lower Triassic of North China, and its implication for lifestyle and environments after the end-Permian extinction 

Dear Dr. Han:

I'm pleased to inform you that your manuscript has been deemed suitable for publication in PLOS ONE. Congratulations! Your manuscript is now with our production department. 

Kind regards, 

on behalf of

Prof. Jörg Fröbisch 

Academic Editor

PLOS ONE